EMBO
reports

# Plasmodium transcription repressor AP2-O3 regulates sex-specific identity of gene expression in female gametocytes

Zhenkui Li[†], Huiting Cui[†], Jiepeng Guan, Cong Liu, Zhengang Yang & Jing Yuan[*] iD

## Abstract

**Male and female gametocytes are sexual precursor cells essential for mosquito transmission of malaria parasite. Differentiation of gametocytes into fertile gametes (known as gametogenesis) relies on the gender-specific transcription program. How the parasites establish distinct repertoires of transcription in the male and female gametocytes remains largely unknown. Here, we report that an Apetala2 family transcription factor AP2-O3 operates as a transcription repressor in the female gametocytes. AP2-O3 is specifically expressed in the female gametocytes. AP2-O3-deficient parasites produce apparently normal female gametocytes. Nevertheless, these gametocytes fail to differentiate into fully fertile female gametes, leading to developmental arrest in fertilization and early development post-fertilization. AP2-O3 disruption causes massive upregulation of transcriptionally dormant male genes and simultaneously downregulation of highly transcribed female genes in the female gametocytes. AP2-O3 targets a substantial proportion of the male genes by recognizing an 8-base DNA motif. In addition, the maternal AP2-O3 is removed after fertilization, which is required for the zygote to ookinete development. Therefore, the global transcriptional repression of the male genes in the female gametocytes is required for safeguarding female-specific transcriptome and essential for the mosquito transmission of *Plasmodium*.**

**Keywords** gametocyte; gametogenesis; gender-specific; *Plasmodium*; transcription repressor
**Subject Categories** Development; Microbiology, Virology & Host Pathogen Interaction

## Introduction

Malaria, caused by the protozoan parasites of the genus *Plasmodium*, is a worldwide infectious disease causing 219 million cases and 430 thousand deaths in 2018 (WHO, 2019). Transmission of

malaria is strictly dependent on the female *Anopheles* mosquitoes. In mammal hosts, the parasites first undergo asexual multiplication in the hepatocytes and then in the erythrocytes. Sexual development starts with a small proportion of intra-erythrocyte asexual parasites irreversibly differentiating into female and male gametocytes, the sexual precursor cells essential for mosquito transmission (Baker, 2010). Within 10–15 min after being ingested into the mosquito midgut, the gametocytes differentiate into gametes and egress from the residing erythrocytes, a process known as gametogenesis. While each female gametocyte produces a single spherical female gamete, a male gametocyte undergoes 3 rounds of DNA replication and mitotic division, giving rise to 8 intracytoplasmic axonemes and subsequently 8 flagellated male gametes (Guttery *et al*, 2015). After being released from the erythrocytes via sequential rupture of the parasitophorous vacuole membrane and the erythrocyte membrane, male and female gametes fertilize to zygotes which further differentiate into crescent-shaped motile ookinetes within 10–20 h. The ookinetes traverse the mosquito midgut and transform to oocysts, each containing thousands of sporozoites (Bennink *et al*, 2016). When the mosquitoes bite again, the sporozoites in the salivary glands are injected into a new mammalian host, which completes the life cycle of malaria parasite.

Detailed transcriptome and proteome studies have revealed that the mature haploid male and female gametocytes exhibit distinct gene and protein expression profiles in the absence of sex chromosomes (Khan *et al*, 2005; Lasonder *et al*, 2016; Miao *et al*, 2017; Yeoh *et al*, 2017; Walzer *et al*, 2018). Compared to asexual blood stage, 2,693 genes (from the total 5,067 genes analyzed in the *P. berghei* genome) are upregulated in the male gametocytes. In contrast, only 1,020 gene expression is augmented in the female gametocytes (Yeoh *et al*, 2017), suggesting dramatic silence of genome-wide transcription in female gametocytes. Gene expression in the gametocytes is in a gender-specific manner. As the male gametocytes undergo three rounds of genome replication and produce 8 highly motile flagellum-like male gametes (Guttery *et al*, 2015; Bennink *et al*, 2016), the transcripts in male gametocytes are highly associated with these cellular events. Despite apparent dormancy of the female gametocytes, ribosome and endoplasmic reticulum in these cells are of great abundance (Khan *et al*, 2005),

---

State Key Laboratory of Cellular Stress Biology, Innovation Center for Cell Signal Network, School of Life Sciences, Xiamen University, Xiamen, China
*Corresponding author. Tel: +86 592 2181601; E-mail: yuanjing@xmu.edu.cn
[†]These authors contributed equally to this work

which is presumably in place for active protein synthesis required for subsequent development. Moreover, a large pool of female-specific mRNA is bound by a conserved translation repression protein complex consisting of DOZI (development of zygote inhibited) and CITH (CAR1/Trailer Hitch homolog) in the female gametocytes (Mair *et al*, 2006; Mair *et al*, 2010). These translationally repressed mRNA in storage will not be translated until needed to fulfill distinct functions in the subsequent development, although the underlying mechanisms remain unknown (Mair *et al*, 2006; Rios & Lindner, 2019). Therefore, these studies provided evidence that male and female gametocytes establish distinct repertoires of gene expression to achieve gender-specific gametogenesis, but failed to uncover how the sex-specific gene expression is controlled.

The ApiAP2 family, including 27 members in the *P. falciparum* and 26 members in the rodent *Plasmodium*, is the largest group of transcription factors (TFs) in malaria parasites (Balaji *et al*, 2005; Campbell *et al*, 2010), many of which have been implicated to govern stage-specific gene transcription during the life cycle of parasites (Yuda *et al*, 2009; Yuda *et al*, 2010; Iwanaga *et al*, 2012; Kafsack *et al*, 2014; Sinha *et al*, 2014; Kaneko *et al*, 2015; Yuda *et al*, 2015; Santos *et al*, 2017). Recently, Modrzynska *et al* and our group systematically investigated the functions of ApiAP2 TFs in the rodent malaria parasite *P. berghei* and *P. yoelii*, respectively (Modrzynska *et al*, 2017; Zhang *et al*, 2017b). Both studies revealed that AP2-O3, an ApiAP2 member, is expressed in the gametocytes and essential for the ookinete formation and mosquito transmission (Modrzynska *et al*, 2017; Zhang *et al*, 2017b). However, the underlying mechanism is unknown. In this study, we established AP2-O3 as a transcription repressor of the male-associated genes in the female gametocytes, which is essential for maintaining the female-specific transcriptome.

# Results

## AP2-O3 is expressed in the female gametocytes and mature oocysts

To define the expression pattern of AP2-O3 in the parasites at different developmental stages, we tagged the endogenous loci of *ap2-o3* with a sextuple HA epitope (6HA) at both the amino (N)- and carboxyl (C)-terminus in the *P. yoelii* 17XNL strain by double crossover homologous replacement using CRISPR/Cas9 method (Zhang *et al*, 2014; Zhang *et al*, 2017a). The resulting strains were referred to as *6HA::ap2-o3* and *ap2-o3::6HA*, respectively (Appendix Fig S1A, B and G). Immunofluorescence assay (IFA) of these two strains showed that AP2-O3 is expressed in the gametocytes and mature oocysts, but not at asexual blood stages or any other stages such as ookinetes, early stage oocysts, and salivary gland sporozoites in mosquito (Fig EV1A). This observation was independently confirmed using a different strain, in which endogenous AP2-O3 was tagged with a red fluorescent protein-mScarlet (Fig EV1B). Consistently, immunoblot analyses also affirmed the restricted expression of AP2-O3 in the gametocytes (Fig EV1C). To dissect whether the expression of AP-O3 exhibits gender specificity, purified gametocytes from both *6HA::ap2-o3* and *ap2-o3::6HA* stains were immunostained with antibody against α-Tubulin (a marker for the male gametocytes) and HA tag. The results showed that AP2-O3

was exclusively present in the female but not male gametocytes (Fig EV1D). Additionally, we tagged AP2-O3 with 6HA in the *P. yoelii* reporter strain *ccp2::mCherry*, in which mCherry was engineered to be specifically expressed in the female gametocytes (Liu *et al*, 2018). As expected, fluorescent signal representing AP2-O3 was only detected in the mCherry⁺ gametocytes (Fig EV1E). In agreement with its role as a putative TF, AP2-O3 was detected only in the nucleus.

## AP2-O3 is essential for ookinete formation and mosquito transmission

To investigate the role of AP2-O3 in the mosquito transmission, we generated two independent mutant strains by deleting the entire coding sequence (2,313 bp) of *ap2-o3* gene in the 17XNL wild-type (WT) and *ap2-o3::6HA* parasites, respectively (Fig 1A). Successful deletion was confirmed by PCR (Appendix Fig S1C and G) and immunoblotting (Fig 1B). The resulting mutants Δ*ap2-o3* and *ap2-o3::6HA;*Δ*ap2-o3* displayed normal asexual blood stages and gametocyte formation in mice (Fig 1C). Moreover, the morphology of purified Δ*ap2-o3* gametocytes was indistinguishable from that of the WT (Fig 1D). However, AP2-O3 disruption blocked the formation of mature ookinetes *in vitro* (Fig 1E), consistent with previous report (Modrzynska *et al*, 2017; Zhang *et al*, 2017b). We isolated the blood bolus from the midguts of the parasite-infected mosquitoes 8 h post-infection (pi) and also failed to detect any mature ookinetes (Fig 1F). Both mutant parasites produced no oocysts in the midgut (Fig 1G) or sporozoites in the salivary glands of the infected mosquitoes (Fig 1H).

To determine whether the above-described phenotype is indeed caused by *ap2-o3* deficiency, we reintroduced a sequence consisting of the coding region of *P. yoelii ap2-o3* (*Pyap2-o3*) and a quadruple Myc epitope (4Myc) back to the *ap2-o3* locus in the Δ*ap2-o3* mutant, generating the complemented strain *com1* (Fig 1I, Appendix Fig S1D and G). The AP2-O3::4Myc was expressed in the gametocytes (Fig 1J) and localized in the nucleus of the female gametocytes (Fig 1K). Importantly, the *com1* parasites produced mature ookinetes *in vitro* (Fig 1L) and midgut oocysts in mosquito (Fig 1M) to a similar level as that of the WT, suggesting that the defects observed in the Δ*ap2-o3* parasites were due to the disruption of AP2-O3. The amino acid sequences of AP-O3, especially in the AP2 (DNA binding) and ACDC domains, are conserved among the *P. yoelii*, *P. berghei*, and *P. falciparum* (Appendix Fig S2). Indeed, complementation of Δ*ap2-o3* with the *P. falciparum ap2-o3* (*Pfap2-o3*) restored ookinete maturation *in vitro* (Fig 1L) and oocyst production in the mosquito midguts (Fig 1M). Together, these results indicate that AP2-O3 has a conserved role in promoting ookinete formation and mosquito transmission.

## AP2-O3 null female gametocytes fail to develop into mature fertile gametes

AP2-O3-deficient parasites produce normal gametocytes but not ookinetes, suggesting a defect in the process of gametocyte–gamete–zygote–ookinete development. Therefore, we sought to delineate the defect in greater details. In line with its expression profile, disruption of AP2-O3 had no appreciable impact on exflagellation of male gametocyte *in vitro* (Fig 2A). Protein expression analysis of

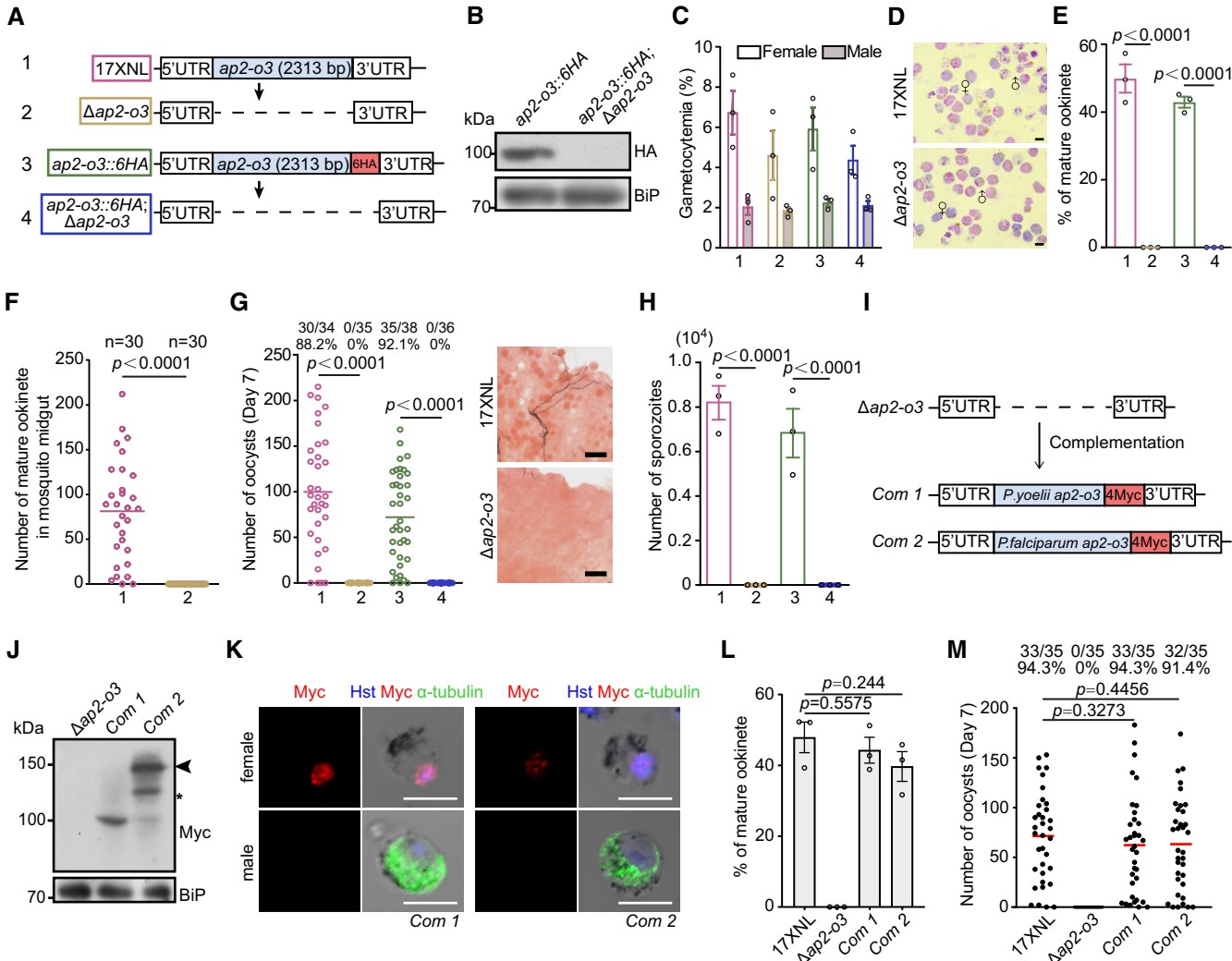

**Figure 1. AP2-O3 is essential for ookinete formation and mosquito transmission.**

A  Diagram depicting CRISPR/Cas9-mediated deletion of the full coding sequence of *ap2-o3* in the 17XNL and *ap2-o3::6HA* strains, respectively, generating the *Δap2-o3* and *ap2-o3::6HA;Δap2-o3* mutants.

B  Immunoblot analyses the AP2-O3 expression in gametocytes of the *ap2-o3::6HA* and *ap2-o3::6HA;Δap2-o3* strains. BiP was used as the loading control.

C  Female and male gametocyte formation in mouse. Gametocytes were counted via Giemsa staining of thin blood smears. Gametocytemia was calculated as the ratio of male or female gametocytes over parasitized erythrocytes.

D  Representative images of purified female and male gametocytes after Giemsa staining. Scale bars = 5 μm.

E  Ookinete maturation *in vitro*. After 12 h of culture, the ookinetes were Giemsa-stained and analyzed for ookinete morphology. The ookinete conversion rate was calculated as the number of mature ookinetes per 100 female gametocytes.

F  Number of ookinetes in the mosquito midguts. The midguts were dissected at 8 h post-blood-feeding and stained with P28 antibody to visualize the crescent-shaped mature ookinete. *n* represents the number of mosquitoes dissected.

G  Oocyst counts in the mosquitoes at day 7 post-blood-feeding. x/y on the top is the number of mosquitoes containing oocyst/the number of mosquitoes dissected; the percentage number is the mosquito infection prevalence. Right panels are the stained midgut oocysts. Scale bars = 50 μm.

H  Salivary gland sporozoite counts in the mosquitoes at day 14 post-blood-feeding.

I  Diagram of CRISPR/Cas9-mediated gene complementation in the *Δap2-o3* mutant. The coding sequence of *ap2-o3* from *P. yoelii* and *P. falciparum* was tagged with a quadruple Myc epitope (4Myc) and introduced back to the *ap2-o3* locus, generating the *Com1* and *Com2* strains.

J  Western blot of the AP2-O3 expression in gametocytes of the *Δap2-o3*, *Com1*, and *Com2*. Protein bands with expected molecular weight (arrows) and an unspecific band (star) were shown.

K  Co-staining of AP2-O3 and α-Tubulin (male gametocyte specific) in gametocytes of *Com1* and *Com2* strains. Scale bars = 5 μm.

L  Ookinete formation *in vitro*. After 12 h of culture, the ookinetes were Giemsa-stained and analyzed for ookinete morphology. The ookinete conversion rate was calculated as the number of mature ookinetes per 100 female gametocytes.

M  Oocyst counts in the mosquitoes at day 7 post-blood-feeding. x/y on the top is the number of mosquitoes containing oocyst/the number of mosquitoes dissected; the percentage number is the mosquito infection prevalence.

Data information: In (A, C, E–G, and H), different colored framelines represent four parasite strains, 17XNL in red, *Δap2-o3* in yellow, *ap2-o3::6HA* in green, and *ap2-o3::6HA;Δap2-o3* in blue. mean ± SEM from three infected mice or independent experiments (C, E, H, and L). Two-tailed unpaired Student's *t*-test applied in (E, H, and L), and Mann–Whitney test applied in (F, G, and M).

TER119 (mouse erythrocyte plasma membrane protein) and SEP1 (parasitophorous vacuole membrane protein) showed proper rupture of these membrane structures of the activated AP2-O3 null gametocytes (Appendix Fig S3A and B). To assess the formation of the female gametes, we took advantage of the fact that protein P28 is translationally repressed in the female gametocytes transiently but de-repressed in the female gametes, zygotes, and ookinetes (Mair *et al*, 2006; Mair *et al*, 2010). Intriguingly, the abundance of P28 was markedly lower in the mutant gametocytes compared to the WT controls 3 h post-xanthurenic acid (XA) stimulation (Fig 2B), which suggest likely compromised fertility of these mutant female gametes. To test this hypothesis, we examined the expression pattern of a marker protein of zygote inner membrane complex (IMC), GAP45, which is apically localized only in the zygotes after fertilization (Wang *et al*, 2020). Two hours after XA stimulation, 84% of the WT female gametes (P28$^+$ only) fertilized to zygotes (P28$^+$/GAP45$^+$), while only 15% of the $\Delta ap2$-$o3$ female gametes achieved to do so (Fig 2C). As a control, parasites with disruption of *cdpk4*, a gene essential for the male gamete formation, produced no zygotes (Billker *et al*, 2004; Jiang *et al*, 2020) (Fig 2C). These results indicate that AP2-O3 disruption also impairs the fertilization ability of the female gametes. Further analyses revealed that the fertilized zygotes of the $\Delta ap2$-$o3$ mutants were developmentally arrested at the early stages (stage II-III), displaying incomplete elongation compared to the WT counterparts (Fig 2D and E). Lastly, we performed genetic crosses between $\Delta ap2$-$o3$ and the male-deficient strain $\Delta map2$ or the female-deficient strain $\Delta nek4$. As expected, the cross between $\Delta map2$ and $\Delta nek4$ produced oocysts that are comparable with the WT. Importantly, the oocyst formation was restored in the $\Delta ap2$-$o3$ parasites that were crossed with the $\Delta map2$ but not the $\Delta nek4$ parasites, further confirming the female-inherited nature of AP2-O3 function (Fig 2F). These results indicate that AP2-O3 null female gametocytes fail to develop into fully functional female gametes for subsequent fertilization and ookinete development (Fig 2G).

### Transcriptome analysis of the female gametocytes

We performed RNA sequencing (RNA-seq) to gain insights into the genome-wide gene expression changes due to the loss of AP2-O3 in the female gametocytes. To obtain female gametocytes of high purity, we used a double fluorescence reporter strain *P. yoelii DFsc7*, in which GFP and mCherry are specifically expressed in the male and female gametocytes, respectively (Liu *et al*, 2018). We deleted the endogenous *ap2-o3* in the *DFsc7* background and generated the mutant *DFsc7*;$\Delta ap2$-$o3$ (Appendix Fig S1G). Of note, the *DFsc7*;$\Delta ap2$-$o3$ strain displayed similar phenotype as was observed in the $\Delta ap2$-$o3$, indicating that introducing the reporter did not interfere with the parasite development (Fig EV2A–D). Using fluorescence-activated cell sorting (FACS), the mCherry$^+$ female gametocytes from both *DFsc7* and *DFsc7*;$\Delta ap2$-$o3$ strains were collected with more than 99% purity for RNA-seq (Fig EV2E and F). Correlation analysis of global gene expression showed good reproducibility among biological replicates (Fig EV2G). Differential gene expression analyses identified 1,141 upregulated genes and 136 downregulated genes (> 2-fold) in the female gametocytes of *DFsc7*;$\Delta ap2$-$o3$ relative to the *DFsc7* parental strain (FDR < 0.05) (Fig 3A). To test whether these differentially expressed genes (DEGs) are sex-related, we

compared them with reported sex-specific or sex-preferential genes in the gametocytes of *P. berghei* (Yeoh *et al*, 2017) and sex-specific transcriptome in the gametocytes of *P. yoelii* from our laboratory. Notably, 53% (606/1,141) of the upregulated DEGs are expressed specifically or preferentially in the male gametocytes. On the other hand, 65% (88/136) of the downregulated DEGs are genes specifically or preferentially expressed in the female gametocytes (Fig 3B). These results indicate that AP2-O3 disruption caused aberrant expression of the sex-specific genes in the female gametocytes. We speculate that AP2-O3 may function as a transcription repressor for the male-related genes to maintain the sex identity of transcriptome in the female gametocytes.

### AP2-O3 represses male gene transcription in the female gametocytes

To understand the biological functions of the upregulated genes, we performed gene ontology enrichment analyses. These genes are mainly enriched in several biological processes, including DNA replication, DNA repair, microtubule-based process, and glycolytic metabolism (Fig 3C). The orthologues of these genes in the *P. berghei* and *P. falciparum* are highly expressed in the male gametocytes but exhibit minimal or no transcriptional activity in the female gametocytes (Yeoh *et al*, 2017). Among those upregulated genes, 26 genes are implicated in the DNA replication and repair pathways (Fig 3D). Increase in the mRNA level of these 26 genes due to AP2-O3 disruption was validated using quantitative real-time–PCR (qRT–PCR) (Fig 3E). Notably, the mRNA levels of these genes determined via RNA-seq and qRT–PCR were positively correlated ($R^2 = 0.68$) (Fig 3F), further confirming the validity of our RNA-seq data. In addition, 34 genes encoding the structural components of flagellum (including dynein heavy chain, dynein intermediate chain, dynein light chain, and centrin) displayed higher expression levels in the female gametocytes of *DFsc7*;$\Delta ap2$-$o3$ compared to *DFsc7* (Fig EV3A and B).

### Protein expression of upregulated male genes in the AP2-O3 null female gametocytes

We next sought to ask whether the upregulated mRNA transcripts of the male-specific genes in the AP2-O3 null female gametocytes could be translated to proteins. From the 26 upregulated genes implicated in the DNA replication and repair pathways, we selected 4: *dpod2* (PY17X_0408600, DNA polymerase delta small subunit), *dpod1* (PY17X_0502300, DNA polymerase delta catalytic subunit), *rpa1* (PY17X_0419400, replication protein A1 small fragment), and *mcm7* (PY17X_0805800, DNA replication licensing factor) (Fig 4A). Each of these 4 genes was endogenously tagged with a *gfp*-coding sequence at the C-terminus in the female-identity reporter strain *ccp2::mCherry*, yielding 4 double-tagged strains, including *ccp2::mCherry;dpod2::gfp* (DTS1), *ccp2::mCherry;dpod1::gfp* (DTS2), *ccp2::mCerry;rpa1::gfp* (DTS3), and *ccp2::mCherry;mcm7::gfp* (DTS4) (Fig 4B). As expected, all the 4 GFP-fusion proteins were expressed and nuclear localized in the male gametocytes (mCherry$^-$) (Fig 4C, E, G and I, upper right panels), but not were detectable (Dpod2::GFP and Dpod1::GFP) or present at extremely low abundance (Rpa1::GFP and Mcm7::GFP) in the female gametocytes (mCherry$^+$) (Fig 4C, E, G and I, upper left panels). Next, we removed *ap2-o3* gene in each

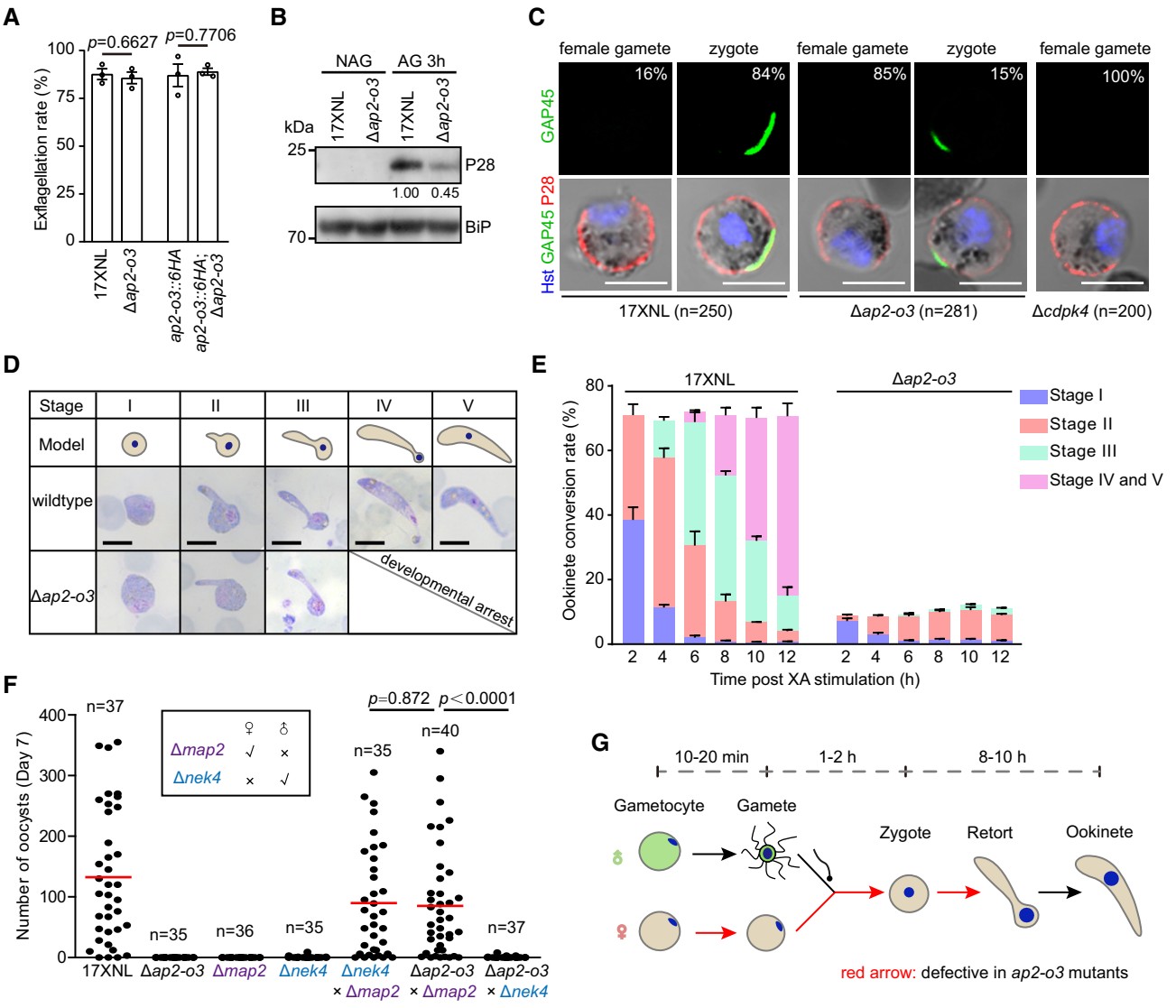

**Figure 2. AP2-O3 null female gametocyte fails to develop fully formed female gamete.**

A *In vitro* exflagellation rate of male gametocytes indicating male gamete formation. mean ± SEM from three experiments. Two-tailed unpaired Student's *t*-test.

B Western blot of P28 protein expression in the non-activated gametocyte (NAG) and activated gametocyte (AG) of 17XNL and Δ*ap2-o3* strains. BiP as loading control. The numbers are the relative intensities of P28 band normalized to the BiP band in the blot.

C Co-staining of P28 and GAP45 (IMC protein) in female gametes (P28 + only) and zygotes (P28$^+$/GAP45$^+$) of the 17XNL, Δ*ap2-o3*, and Δ*cdpk4* strains. Percentages of female gametes and zygotes were indicated, respectively. *n* is the number of cells counted. Scale bars = 5 μm. Note: The signal of P28 and GAP45 is reduced similarly in all Δ*ap2-o3* female gametes and zygotes compared to that of 17XNL.

D Time-course analysis of zygote to ookinete development *in vitro*. Upper diagrams indicate morphological changes from zygote to ookinete. Scale bars = 5 μm.

E Quantitative counting of different stages during parasite development in (D). Data are gathered from 3 independent experiments and presented as mean ± SEM.

F Day 7 midgut oocyst counts from mosquitoes infected with parasites, including 17XNL, Δ*ap2-o3*, Δ*map2*, or Δ*nek4* strain alone as well as mixtures of Δ*map2*/Δ*nek4*, Δ*ap2-o3*/Δ*map2*, and Δ*ap2-o3*/Δ*nek4*. Δ*nek4* and Δ*map2* are female and male gamete-defect parasites, respectively. *n* is the number of mosquitoes dissected, Mann–Whitney test applied, and two experiments repeated.

G Schematic of the defects caused by AP2-O3 deficiency during female gametocyte to ookinete development.

individual *DTS* strain (Appendix Fig S1G). As expected, AP2-O3 disruption did not significantly affect the level of any of these 4 proteins in the male gametocytes as revealed by both fluorescence microscopy and flow cytometry (Fig 4D, F, H and J). In clear contrast, protein levels of Dpod2::GFP and Dpod1::GFP in the female gametocytes were dramatically increased (Fig 4D and F),

while Rpa1::GFP and Mcm7::GFP levels in the female gametocytes showed a modest increase (Fig 4H and J) compared to the corresponding parental strains. Interestingly, the protein levels of these 4 genes in the female gametocytes were still much lower than that in the male gametocytes counterpart of the same AP2-O3-deficient parasites (Fig 4D, F, H and J).

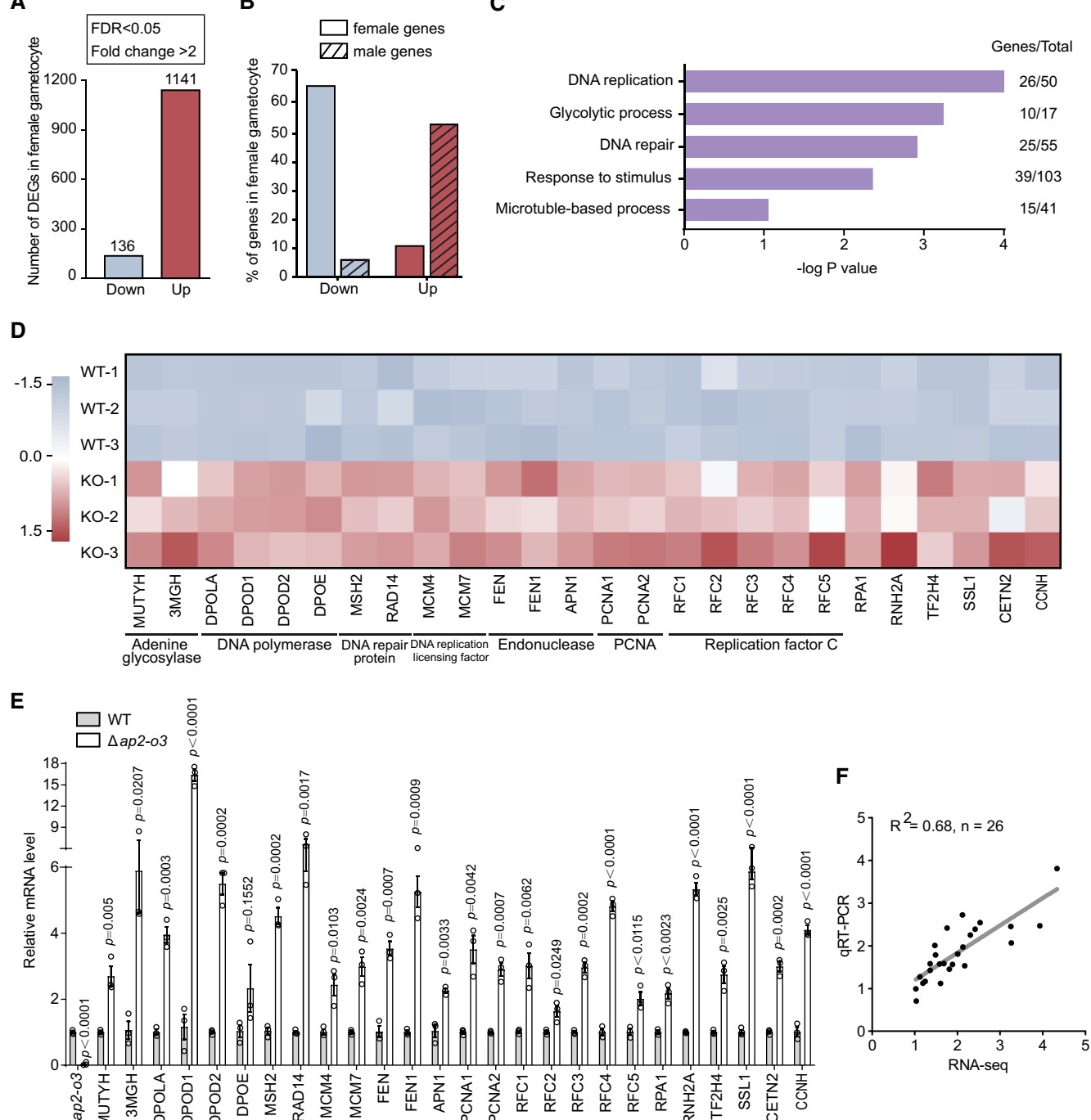

**Figure 3. AP2-O3 represses transcription of the male-associated genes in the female gametocyte.**

A  Number of differentially expressed genes in the purified female gametocyte determined by RNA-seq. Detailed information is available in Fig EV2.
B  Percentage of female and male genes in down- and upregulated genes.
C  Gene ontology enrichment analysis of the upregulated genes indicates male-specific or preferential biological processes.
D  Transcript expression heatmap by RNA-seq of 26 genes involving DNA replication and DNA repair between *DFsc7* (WT) and *DFsc7;Δap2-o3* (KO) strains.
E  Quantitative RT–PCR analysis of 26 genes in (D). mean ± SEM from three independent experiments. Two-tailed unpaired Student's *t*-test applied.
F  Linear correlation of gene expression among the 26 genes (DNA replication and DNA repair) detected via RNA-seq and quantitative RT–PCR.

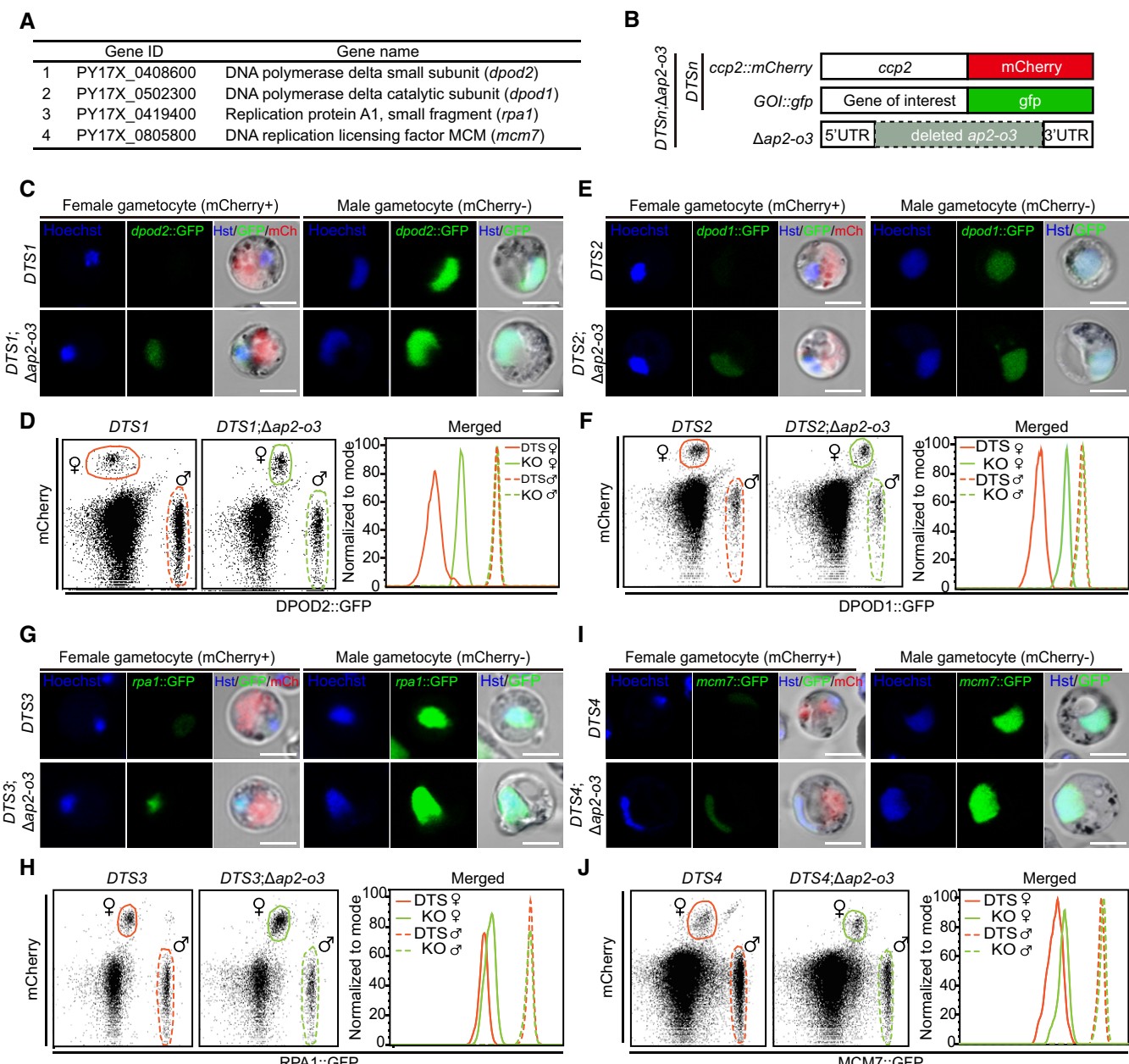

**Figure 4. Protein expression of upregulated male genes in AP2-O3 null female gametocytes.**

A Four male genes in DNA replication and DNA repair pathways, including *dpod2*, *dpod1*, *rpa1*, and *mcm7*.

B Diagram of CRISPR/Cas9-mediated C-terminally tagging of these four genes with *gfp* in the female reporter strain *ccp2::mCherry*, generating four *DTS* (double-tagged strain: *DTS1-DTS4*). Next, the endogenous *ap2-o3* gene was deleted in these four *DTS*, generating four *DTS;Δap2-o3* mutants.

C Representative fluorescence microscopy images of mCherry and DPOD2::GFP expression in female and male gametocytes of the *DTS1* (*ccp2::mCherry;dpod2::gfp*) and *DTS1;Δap2-o3* strains. mCherry is specifically expressed in female gametocytes. Scale bars = 5 μm.

D Flow cytometry detection of mCherry and DPOD2::GFP expression in female and male gametocytes of the *DTS1* and *DTS1;Δap2-o3* strains. The female and male gametocyte populations are circled by solid and dashed line, respectively.

E, F Similar analysis (as in C, D) in *DTS2* (*ccp2::mCherry;dpod1::gfp*) and *DTS2;Δap2-o3* strains. Scale bars = 5 μm.

G, H Similar analysis (as in C, D) in *DTS3* (*ccp2::mCherry;rpa1::gfp*) and *DTS3;Δap2-o3* strains. Scale bars = 5 μm.

I, J Similar analysis (as in C, D) in *DTS4* (*ccp2::mCherry;mcm7::gfp*) and *DTS4;Δap2-o3* strains. Scale bars = 5 μm.

## AP2-O3 binds to the upstream promoter of specific male genes

To determine the DNA binding sites and direct gene targets of AP2-O3, we performed 2 independent chromatin immunoprecipitation

coupled with high-throughput sequencing (ChIP-seq) experiments using the gametocytes isolated from the *ap2-o3::6HA* parasites. Using $P$-value $< 1e^{-5}$ and fold enrichment $> 2$ as the cut-off, 1,303 peaks (belonging to 1,199 genes) were identified in the 1[st] experiment and

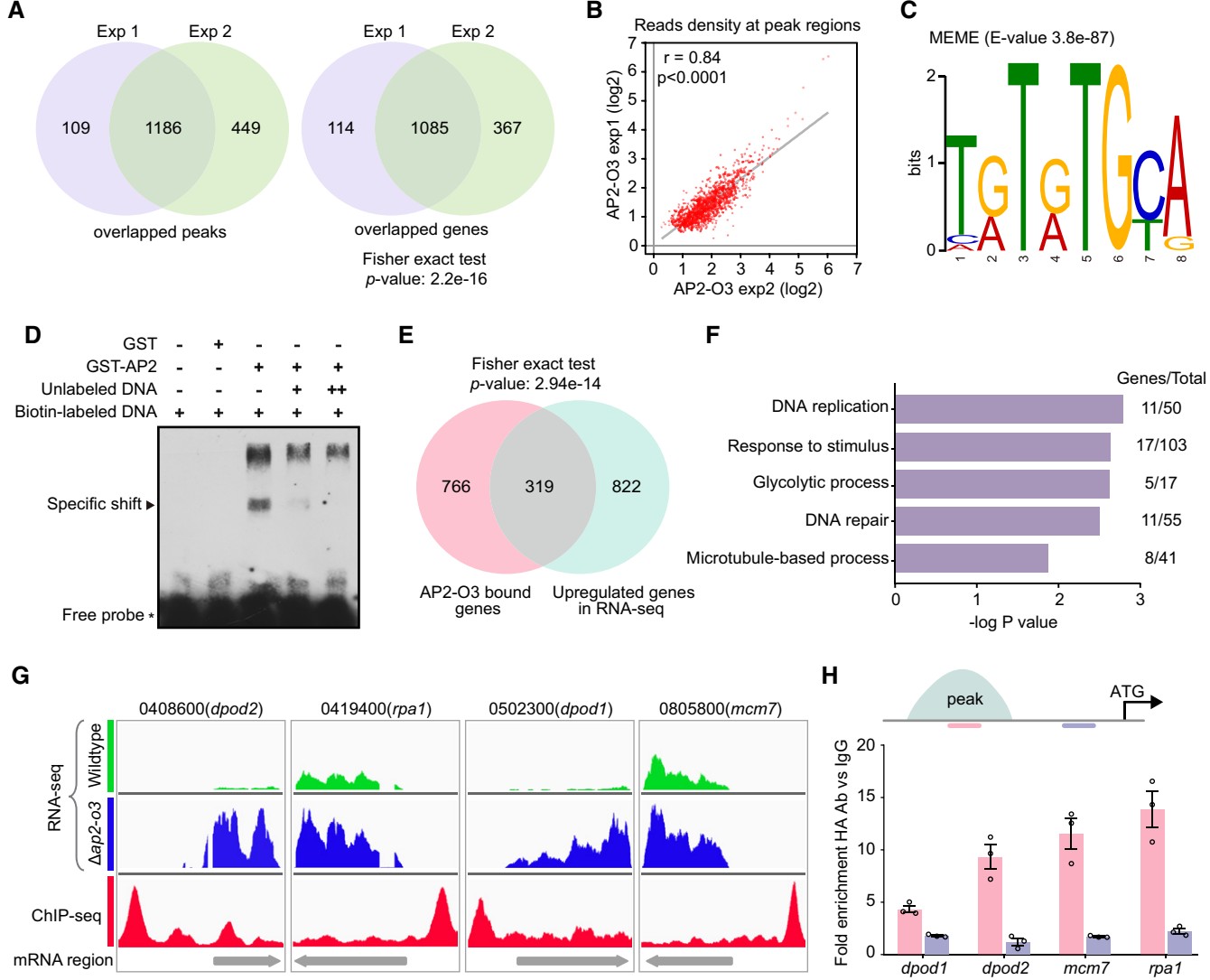

**Figure 5. AP2-O3 binds upstream promoter of specific male genes.**

A   The number of overlapped peaks (1,186) and genes (1,085) in two ChIP-seq experiments. Statistical differences were determined using Fisher exact test.
B   Correlation of read density among the 1,186 peaks between two ChIP-seq experiments. Pearson correlation is indicated.
C   The enriched DNA motif is identified by the motif-discovery algorithm MEME in AP2-O3 binding sites.
D   EMSA using the recombinant GST-fused AP2 domain of AP2-O3 and a synthesized DNA probe containing three repeats of predicted motif. GST was used as a negative control. An arrowhead indicates the shifted band.
E   The number (319) of overlapping genes between AP2-O3-bound genes (1,085) in ChIP-seq and upregulated genes (1,141) in RNA-seq. Statistical differences were determined using Fisher exact test.
F   Gene ontology enrichment analysis of the 319 genes indicates male-specific or preferential biological processes. Hypergeometric test was applied.
G   Mapped views showing the RNA-seq and ChIP-seq results of four male genes in DNA replication and DNA repair pathways.
H   ChIP-qPCR validation of the binding between AP2-O3 and the upstream region of four male genes in (G). The regions detected via ChIP-qPCR are indicated by pink line (in the peak) and purple line (out of the peak). mean ± SEM from three independent experiments.

1,657 peaks (belonging to 1,452 genes) in the 2$^{nd}$, with an overlap of 1,186 peaks (corresponding to 1,085 genes) (Fig 5A). We observed high reproducibility between the two ChIP-seq experiments with a Pearson correlation coefficient of 0.84 (Fig 5B). Analyses with MEME (Bailey & Elkan, 1994) showed that an 8-base sequence, T(G/A)T(G/A)TGCA, was the most frequently concentrated motif around the DNA sequence of the binding peaks (Fig 5C). To determine

whether AP2-O3 binds to this putative motif, we performed electrophoretic mobility shift assay (EMSA) using purified recombinant GST-tagged AP2 domain (DNA binding domain) of AP2-O3 and synthetic nucleotides containing three repeats of the motif sequence (TGTGTGCAta). A striking shift was observed for the biotin-labeled DNA probe in an AP2-O3-dependent manner, and the shift was eliminated competitively by a 200-fold unlabeled DNA probe (Fig 5D).

Among the 1,085 genes recovered from our ChIP-seq experiments, 319 genes were also identified in the RNA-seq analyses described above (Fig 5E). Gene ontology enrichment analyses performed on these 319 genes revealed significant over-representation of biological activities restricted to the males, which include DNA replication, DNA repair, microtubule-based process, and glycolytic process (Fig 5F). In the genome mapped view, clear peaks were observed at the upstream of the coding sequence of certain male-specific genes, including *dpod2, rpa1, dpod1,* and *mcm7* (Fig 5G). The ChIP-qPCR analyses further validated that the peak regions bound with AP2-O3 are located at the upstream of these four genes (Fig 5H). These results demonstrate that AP2-O3 associates with the promoters of given male genes.

To further establish AP2-O3 as a transcription repressor of the male genes, we aimed to ectopically express *ap2-o3* in the male gametocytes (Appendix Fig S1F and G, Fig EV4A). To achieve that, an *ap2-o3::6HA* expressing cassette under the control of the 5′-UTR (1,300 bp) of male gametocyte specific gene *migs* (Tachibana *et al,* 2018) and the 3′-UTR (561 bp) of *dhfr* gene was inserted into the *p230p* locus using CRISPR/Cas9 method (Manzoni *et al,* 2014; Philip & Waters, 2015). This transgenic line is referred to as EE2. A *mCherry*-expressing transgenic line EE1 was generated as the control (Fig EV4A). As expected, *ap2-o3* mRNA was found to be elevated in the EE2 compared to the EE1 parasites (Fig EV4B). Both mCherry and AP2-O3::6HA were ectopically expressed in the male gametocytes (Fig EV4C and D). Importantly, slight but significant decrease in the mRNA levels was observed for the male-specific genes, including *dpod2, rpa1, dpod1,* and *mcm7*, in the purified male gametocytes of EE2 compared to EE1 parasites (Fig EV4E). These results suggest that AP2-O3 suppresses the transcription of these given male genes. However, we analyzed the genome DNA content change during male gametogenesis using the cytometry but detected genome DNA replication indistinguishable in activated male gametocytes between EE1 and EE2 (Fig EV4F).

## Decreased expression of highly expressed female-specific genes in the AP2-O3 null female gametocytes

AP2-O3 depletion causes massive transcriptional activation of numerous male genes that are otherwise dormant in the female gametocytes. However, it is not clear whether this transcriptome dysregulation affects the intrinsic transcription program of the female genes at the same time. Global transcriptome in the WT female gametocytes was visualized by ranking the genes according to the transcript levels determined by RNA-seq (Fig 6A, left panel). The genes with highest scores (FPKM > 1,500) in the WT female gametocytes encode parasite surface proteins (P25 and P28), IMC-associated proteins (GAP45, GAP40, IMC1i, IMC1c, IMC1h, SPM1, SPM2, GAPM1, and GAPM2), glideosome component (Myosin A and MTIP), ookinete-secreted protein (POP), and ookinete crystalloid protein (LCCL) (Fig 6A, right panel), all of which are heavily implicated in the female physiology, such as female gametogenesis, fertilization, ookinete differentiation, ookinete gliding, and midgut traversal of ookinete (Bennink *et al,* 2016). Strikingly, majority of genes with FPKM < 1,500, which accounted for 73% of the coding genes of the parasite genome, exhibited augmented transcripts (Fig 6A). In contrast, 85 (77%) out of the 111 genes with

FPKM > 1,500 were decreased in expression (Fig 6B), although by less than twofold in the female gametocytes of *DFsc7*;Δ*ap2-o3* compared to *DFsc7* parental parasites (Fig 6C). Of these 85 genes, we selected 10 and performed qRT–PCR to confirm their significant downregulation (Fig 6D). Consistently, we detected dramatically decreased protein levels of both P28 and GAP45 in the activated female gametocytes of Δ*ap2-o3* compared to the WT parasites (Figs 2B and 6E).

Among the highly abundant female-specific mRNAs, most of them are known to be stabilized and translationally repressed by DOZI/CITH complex (Mair *et al,* 2006), and genetic ablation of either DOZI or CITH causes degradation of these mRNAs (Mair *et al,* 2010). We asked if the impact of AP2-O3 disruption on female gene expression is through the DOZI/CITH complex. Of the known components of DOZI/CITH complex, 9 genes, including *dozi, cith, eIF4e, pabp1, celf2, alba1, alba2, alba3,* and *phosphoglycerate mutase* (Mair *et al,* 2010), were revealed to be slightly increased (20-30%) at the mRNA level in the female gametocytes of *DFsc7*;Δ*ap2-o3* in comparison to the *DFsc7* parental parasites (Appendix Fig S4A). qRT–PCR showed that the mRNA levels of the *dozi* and *cith* genes remained unaffected in the purified female gametocytes of Δ*ap2-o3* compared to the WT (Appendix Fig S4B). Furthermore, we tagged endogenous *dozi* and *cith* genes with a 6HA epitope in both the 17XNL and Δ*ap2-o3* strains (Appendix Fig S1G). However, no significant changes in the DOZI and CITH expression were detected in the female gametocytes after the AP2-O3 disruption (Appendix Fig S4C and D). These results suggest that AP2-O3 depletion in the female gametocytes does not affect mRNA or protein level of the key components of translation repression complex.

## Downregulation of AP2-O3 expression after fertilization

AP2-O3 is present in the female gametocytes but not ookinetes, indicating that its function is precisely regulated during normal development (Fig EV1A–C). To study the expression dynamics of AP2-O3 in details, we collected *ap2-o3::6HA* parasite at different stages from an *in vitro* culture and performed IFA with antibodies against HA and P28. Compared to the 100% expression in the female gametocytes (Fig EV1C), AP2-O3 is only detected in 23% (46/200) of the female gametes (P28$^+$) 30 min post-XA treatment and in 4% (8/200) of the zygotes 60 min post-XA treatment (Fig 7A). By 2 h, no expression of AP2-O3 was detected in the retorts (0/200) (Fig 7A). Immunoblot analyses also revealed that AP2-O3 protein level diminished in a time-dependent fashion (Fig 7B).

## Ectopic expression of AP2-O3 after fertilization impairs the ookinete development

We speculate that ectopic AP2-O3 expression after fertilization will affect normal zygote to ookinete development. To test it, we attempted to generate a strain ectopically expressing AP2-O3 after the gametogenesis (Fig 7C). Specifically, in the *ap2-o3::6HA* parasites, the 800 bp promoter sequence of *ap2-o3* was replaced with a 1,200 bp promoter of *ccp2*, a gene that is transcribed in the female gametocytes, female gametes, zygotes, and ookinetes (Liu *et al,* 2018) (Fig 7C). Correct replacement in the resulting mutant *Pccp2*

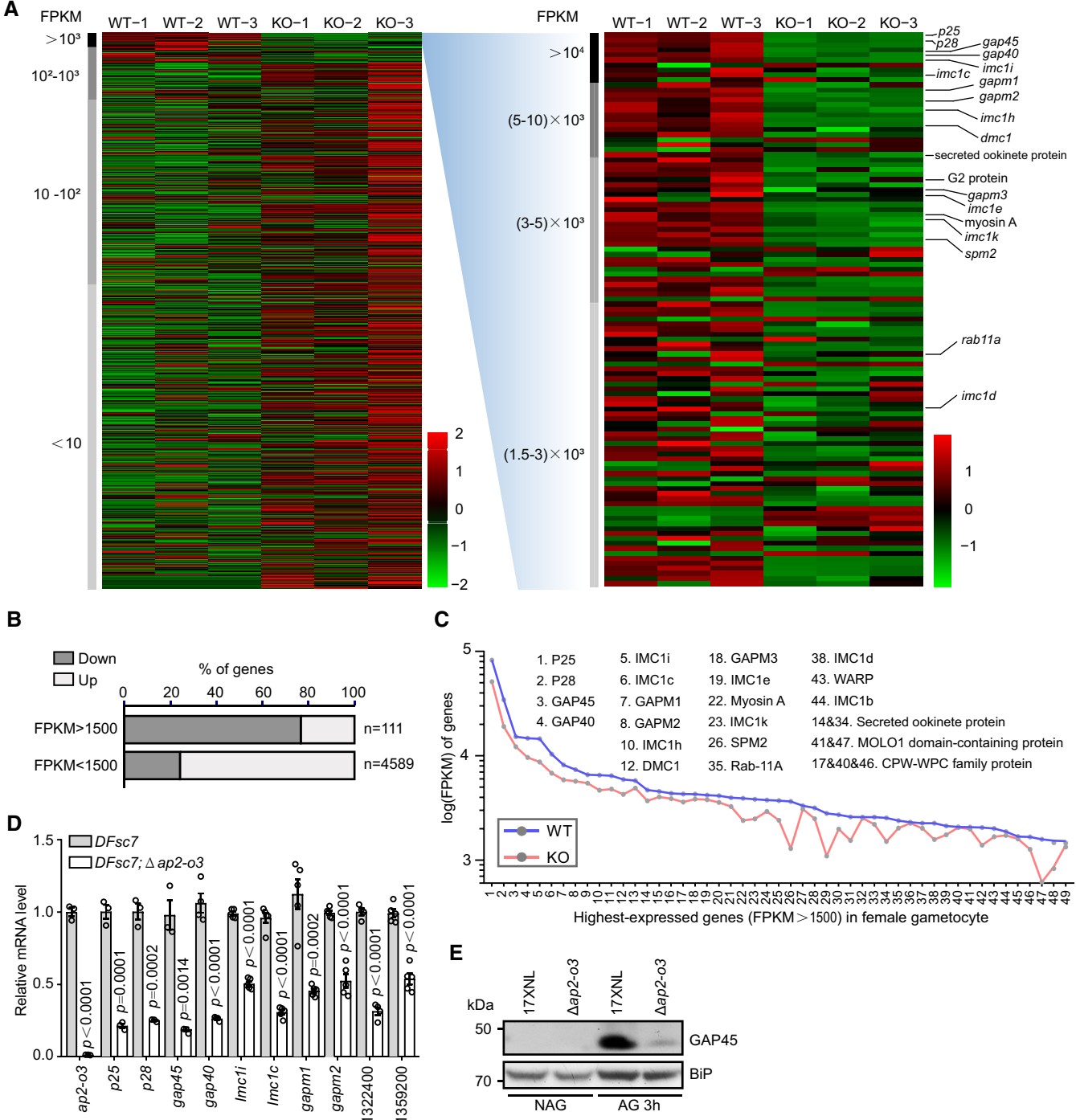

**Figure 6. Decreased expression of highly expressed female genes in AP2-O3 null female gametocytes.**

A  Transcriptome of female gametocyte by RNA-seq between *DFsc7* (WT) and *DFsc7;Δap2-o3* (KO) strains. The genes were ranked by transcripts level (FPKM average of three sample replicates) in the heatmap. In spite of global upregulation of vast genes with low or medium transcript levels (left panel), the highly expressed genes (FPKM > 1,500, right panel) are downregulated in KO over WT. The FPKM values in each row (each gene) were normalized by Z-score normalization.

B  Gene percentage of downregulation and upregulation in populations of the highly expressed genes (*n* = 111, FPKM > 1,500) and the others (*n* = 4,589, FPKM < 1,500) between KO and WT.

C  Line chart of 50 representative highly expressed female genes in (A). The dot is the mean value of FPKM in three repeats.

D  qRT–PCR detecting mRNA transcript level of ten selected highly expressed genes in (A). The numbers indicate the gene ID. mean ± SEM from three to five independent experiments. Two-tailed unpaired Student's *t*-test applied.

E  Western blot of GAP45 protein expression in non-activated (NAG) and activated gametocyte (AG) of 17XNL and *Δap2-o3* strains. BiP as loading control.

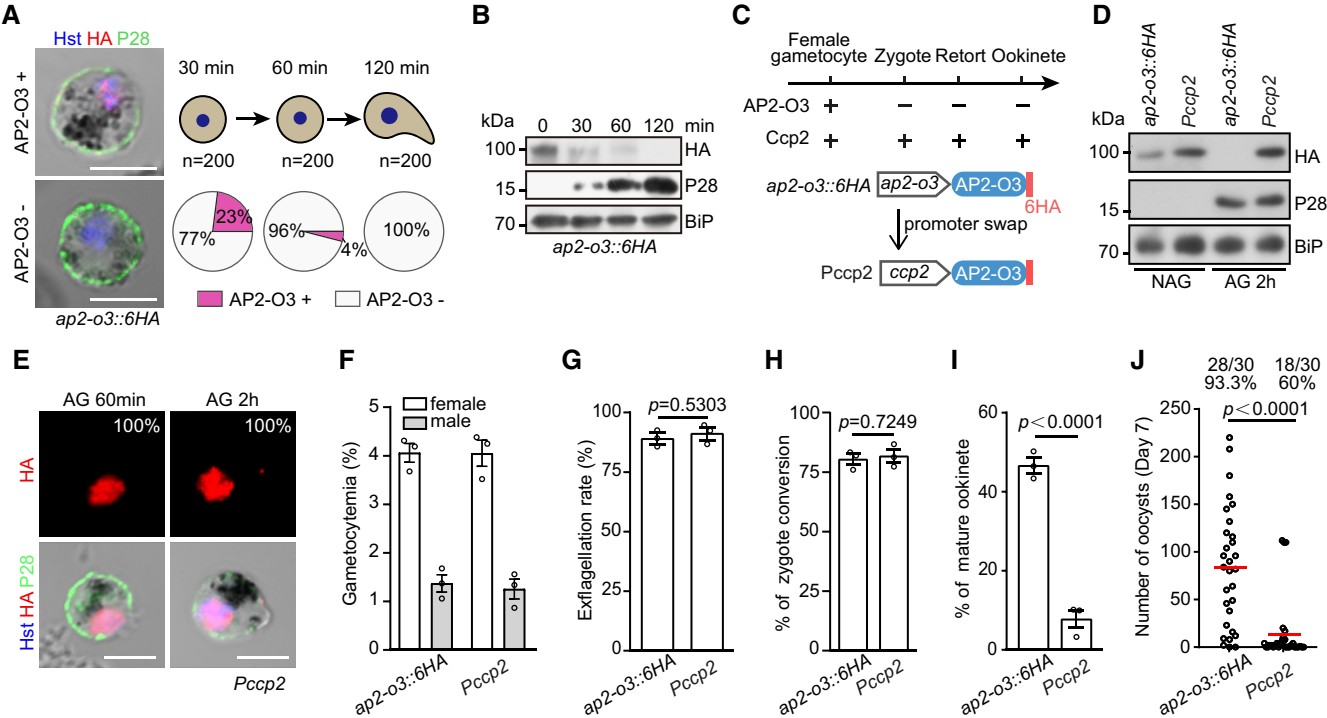

**Figure 7.  Downregulation of AP2-O3 expression after fertilization is required for ookinete development.**

A   AP2-O3 protein expression from female gametocyte, female gamete, zygote, to retort stage of the *ap2-o3::6HA* strain. The activated gametocytes were collected at 30, 60, and 120 min and co-stained with HA and P28 antibodies. Pie charts indicate the percentage of AP2-O3+ and AP2-O3− cells. "*n*" is the number of cells analyzed at each time point. Scale bars = 5 μm.

B   Western blot of AP2-O3 protein expression in parasites from (A).

C   Diagram of CRISPR/Cas9-mediated promoter swap in the *ap2-o3::6HA*. The promoter (467 bp) of *ap2-o3* gene was replaced with the *ccp2* gene promoter (1,202 bp), generating the *Pccp2* mutant to drive gene expression post-fertilization.

D   Western blot of AP2-O3 protein expression in non-activated (NAG) and activated gametocytes (AG) of *ap2-o3*::6HA and *Pccp2* strains.

E   IFA of AP2-O3 protein expression in activated gametocytes of *Pccp2* strains. Scale bars = 5 μm.

F   Gametocyte formation in mouse.

G   *In vitro* exflagellation rate of male gametocytes.

H   Female gamete fertilization to zygote analysis by co-staining of P28 and GAP45.

I    Mature ookinete formation *in vitro*.

J    Oocyst counts in the mosquitoes at day 7 post-blood-feeding. x/y on the top is the number of mosquitoes containing oocyst/the number of mosquitoes dissected; the percentage number is the mosquito infection prevalence.

Data information: In (F–I), mean ± SEM from three infected mice or three experiments. Two-tailed unpaired Student's *t*-test applied in (G-I), and Mann–Whitney test applied in (J).

was confirmed by PCR (Appendix Fig S1E and G), which drove AP2-O3 expression in the non-activated gametocytes at a level comparable to that of the parental *ap2-o3::6HA* parasites (Fig 7D). The AP2-O3 expression was maintained in the activated *Pccp2* gametocytes even at 2 h post-XA stimulation (Fig 7D). IFA analyses also confirmed the AP2-O3 expression in all (93/93) of the activated female gametocytes of the *Pccp2* parasites (Fig 7E). Compared to the parental strain, the *Pccp2* strain produced normal gametocytes in mice (Fig 7F), male gametes *in vitro* (Fig 7G), and fertilized zygotes (Fig 7H). However, the ability of the zygotes to develop into ooki-netes *in vitro* and midgut oocysts in the mosquitoes was signifi-cantly reduced (Fig 7I and J). The detrimental effect imposed on the ookinete development caused by the ectopic expression of AP2-O3 post-fertilization suggests that AP2-O3 expression is precisely regu-lated to meet the developmental demands.

## Sequential expression of AP2-O3 and AP2-O during female gametogenesis

ApiAP2 member AP2-O functions as a transcription activator, and the parasites lacking AP2-O fail to produce mature ookinetes in the *P. yoelii* and *P. berghei* (Yuda *et al*, 2009; Zhang *et al*, 2017b). We sought to explore the potential relation between AP2-O3 and AP2-O. AP2-O was reported to be expressed after gametogenesis but not in non-activated female gametocytes (Yuda *et al*, 2009; Zhang *et al*, 2017b). To study the expression profile of both AP2-O3 and AP2-O in the same parasites, we tagged the endogenous AP2-O with a 4Myc epitope in the *ap2-o3::6HA* strain, generating the doubly tagged strain *ap2-o3::6HA;ap2-o::4Myc* (Appendix Fig S1G). IFA confirmed the expression of AP2-O3 but not AP2-O in the nucleus of the female gametocytes (Appendix Fig S5A). After

gametogenesis, AP2-O started to be expressed in the female gametes, zygotes, retorts, and ookinetes (Appendix Fig S5A), while AP2-O3 expression quickly disappeared (Appendix Fig S5A and Fig 7A). These results are consistent with the *ap2-o* transcript being translationally repressed in the female gametocytes (Yuda *et al*, 2009).

Both RNA-seq and qRT–PCR revealed that the *ap2-o* mRNA level remained unaffected in the female gametocytes of *DFsc7*;Δ*ap2-o3* compared to the *DFsc7* parental parasites (Appendix Fig S5B and C). Furthermore, we deleted *ap2-o3* in the *P. yoelii ap2-o::6HA* strain (Zhang *et al*, 2017b), generating the mutant *ap2-o::6HA*;Δ*ap2-o3* (Appendix Fig S1G). As expected, AP2-O3 depletion had no appreciable effect on the AP2-O expression in the female gametes or zygotes (Appendix Fig S5D). Together, these results demonstrated a sequential expression of AP2-O3 and AP2-O in the parasite development before and after gametogenesis.

## Discussion

*Plasmodium* male and female gametocytes are indispensable sexual precursor cells for parasite transmission in the mosquitoes. Differentiation of gametocytes to fertile gametes relies on the gender-specific gene expression (Guttery *et al*, 2015). How the parasites establish distinct repertoires of transcriptome in the male and female gametocytes remains largely unknown. Here, we report that the TF family ApiAP2 member AP2-O3, specifically active in the female gametocytes, is a transcription repressor that regulates the formation of female gamete, fertilization, and early development post-fertilization. Transcriptome analyses by RNA-seq show that AP2-O3 disruption leads to widespread upregulation of numerous male genes in the female gametocytes. Majority of these upregulated genes are implicated in essential biological processes during male gametogenesis, such as DNA replication, DNA repair, axoneme biogenesis, and flagella assembly. Our results also suggest that AP2-O3 represses the gene expression programmed for male gametogenesis, thereby assisting gametogenesis of the opposing gender. Moreover, RNA-seq reveals that genes with minimum or low transcriptional activities are also moderately upregulated at a global range in the female gametocytes of the AP2-O3 null parasites, raising the possibility that AP2-O3 also plays a role in repressing the transcription of nonessential genes in the female gametocytes.

Interestingly, although the mRNA and protein levels of 4 male-associated genes (*dpod1*, *dpod2*, *rpa1*, and *mcm7*) in the female gametocytes are elevated due to AP2-O3 disruption, they never reach to a comparable degree in the naturally occurring male gametocytes. One possibility is a lack of transcription activators for these male genes in the female gametocytes. Their activation is limited even if they are released from the transcriptional repression imposed by AP2-O3. On the other hand, ApiAP2 TFs are found to associate with transcription regulators or chromatin remodelers such as BDP1, BDP2, and histone acetyltransferase in the *Plasmodium* (LaCount *et al*, 2005; Josling Gabrielle *et al*, 2015; Santos *et al*, 2017; Toenhake *et al*, 2018), and histone deacetylase and acetyltransferase in the *Toxoplasma. gondii* (Saksouk *et al*, 2005; Dixon *et al*, 2010). A transcription regulatory complex composing AP2-O3 and other factors may function together to fulfill transcription repression, which could not be completely lifted solely through

abolishing AP2-O3. Consistent with this speculation, the expression of male genes is only slightly reduced in the transgenic male gametocytes with ectopic expression of AP2-O3.

AP2-O3 null parasites generate female gametocytes, which however fail to develop into fully fertile female gametes for subsequent fertilization and early ookinete development. As a transcription repressor of male-associated genes and/or lowly transcribed genes, it is not clear how does AP2-O3 function in the female gametocytes. In addition to the global upregulation of those moderately transcribed genes (FPKM < 1,500 determined by RNA-seq), downregulation of most of the highly active gene transcripts (FPKM > 1,500) is also observed in the AP2-O3 null female gametocytes. Strikingly, these highly transcribed genes are essential for various biological processes in the females, such as gametogenesis, fertilization, and zygote to ookinete differentiation. Therefore, massive downregulation of their expression likely underlies the defects of the AP2-O3 null parasite. As to the mechanisms how the female genes are downregulated due to AP2-O3 disruption, there are 3 possibilities (see the proposed model in Fig EV5). First, there exists a competition mechanism between the female and male (or non-female) gene expression program in the female gametocytes, which are in a relatively quiescent status in transcription (Yeoh *et al*, 2017; Witmer *et al*, 2020). The global upregulation of male genes expropriates the resource needed for transcription and translation of the female gene expression programs. In this scenario, the purpose of AP2-O3 to inhibit the global transcription of male (or non-female) genes is to safeguard proper expression of the female genes in the female gametocytes. While this is an intriguing hypothesis, currently it is technically challenging to be tested. The second possibility is that in the absence of AP2-O3, male transcripts/proteins that are of super abundance subvert the function of female transcripts/proteins in the female gametocytes. The last possibility is that among the upregulated genes caused by AP2-O3 disruption, unknown factor could repress the female gene expression. Recently, the *P. berghei* TF AP2-FG was identified to be expressed in the female gametocytes. However, it is a transcription activator for genes necessary for female gametocyte maturation (Yuda *et al*, 2020).

AP2-O3 is expressed in the female gametocytes but not ookinetes. Detailed analyses indicated that the expression of AP2-O3 quickly diminished after the gametogenesis and completely disappeared after the fertilization (Fig 7). The short time window of its expression indicates a stage-specific function of AP2-O3 in the female gametocytes. It also raises the question why its expression completely vanished after fertilization. It has been reported that after fertilization, the diploid zygotes undergo an immediate genome duplication (to tetraploid) and establish a subpellicular microtubules network to support cellular morphogenesis, which is essential for the ookinete formation (Guttery *et al*, 2015). Given the ability of AP2-O3 to suppress the transcription of genes implicated in the DNA replication and microtubule-related processes, it is reasonable for the zygotes to eliminate AP2-O3 to meet the developmental demands. Consistent with this speculation, constitutive transgenic expression of AP2-O3 post-fertilization impaired the zygote to ookinete development. Together, the strict spatiotemporal regulation of AP2-O3 coordinates the female gametogenesis and zygote to ookinete development, mimicking the eukaryotic zygote genome activation, in which maternal repressive factors are

required to be removed, allowing the zygotic genome transcription after fertilization (Schulz & Harrison, 2019). Identifying factors that fine-tune the spatiotemporal expression of AP2-O3 at either the transcription or translation level will be of great interest in the future.

# Materials and Methods

### Animal and parasite usage and ethics statement

The ICR mice (female, 5–6 weeks old) were used for parasite propagation, drug selection, parasite cloning, and mosquito feeding. The mice were purchased from the Animal Care Center of Xiamen University. All mouse experiments were performed in accordance with approved protocols (XMULAC20140004) by the Committee for Care and Use of Laboratory Animals of Xiamen University. All transgenic parasites were generated from *P. yoelii* 17XNL strain using the CRISPR/Cas9 method (Zhang *et al*, 2014; Zhang *et al*, 2017a). The parasites were constantly subjected to mosquito transmission for maintaining the gametocyte formation. All the parasite strains generated in this study are listed in Appendix Table S1.

### Plasmid construction

CRISPR/Cas9 plasmid pYCm was used for genomic modification (Zhang *et al*, 2014; Zhang *et al*, 2017a). To construct the plasmids for gene deleting, the 5′- and 3′-flanking genomic sequence (400–700 bp) of target genes was PCR-amplified as left and right homologous arms and inserted into the restriction sites of pYCm. Oligonucleotides for small guide RNAs (sgRNAs) were annealed and ligated into pYCm. Two sgRNAs were designed to target the coding region of each gene. To construct the plasmids for gene tagging, the C- or N-terminal segments (400–800 bp) of the coding regions were PCR-amplified as the left or right arm and 400–800 bp from 5-UTR or 3-UTR following the translation stop codon as left and right arm, respectively. A DNA fragment encoding GFP, mScarlet, 6HA, or 4Myc was inserted between the left and right arms in frame with the gene of interest. For each gene tagging, at least three sgRNAs were designed to target the C- or N-terminal of the coding region. To construct the plasmids for nucleotide replacement, the substitution sites were designed with a restriction site for each detection and placed in the middle of the homologous arms. Mutagenesis was performed using the KOD One™ PCR Master Mix (TOYOBO, KMM-101). All the primers and oligonucleotides used in this study are listed in Appendix Table S2.

### Parasite transfection and gene modification

Plasmid transfections of parasites were performed via electroporation using Nucleofector™ 2B (Lonza). Blood with 10–20% parasitemia collected from infected mice was cultured in RPMI-1640 (Gibco, #11879020) supplied with 20% FBS (Gibco, #10099) for 3 h at 37°C for schizont enrichment. The schizonts were washed two times with RPMI1640 and electroporated with 5 μg circular plasmid DNA using Lonza Nucleofector. Transfected parasites were injected into a naive mouse through the tail vein. Pyrimethamine (7 μg/ml) supplied in drinking water was provided to mice for drug selection

24 h after transfection. Parasites usually appear 5–8 days after drug selection. Genomic DNA of parasites was extracted for genotype PCR analysis. Parasite clones with correct modification were obtained using limiting dilution method. All the primers for genotype PCR are listed in Appendix Table S2.

### Negative selection of modified parasites

Modified parasites subject for sequential modification were negatively selected to remove episome pYCm plasmid. Briefly, 2.0 mg/ml 5-Fluorouracil (5FC, Sigma, F6627) in drinking water was provided to parasite-infected mice for 8 days with new drug replacement on day 4. Complete removal of pYCm plasmid within parasites was monitored by PCR genotyping. To further confirm the loss of plasmid, the negatively selected parasites were subject to a pyrimethamine sensitivity test. Only the successful negatively selected parasites without pYCm could be subject to next round genome editing.

### Gametocyte induction in mouse

To facilitate gametocyte formation, ICR mice were treated with phenylhydrazine (Sangon Biotech, #A600704; 80 μg phenylhydrazine/g mouse body weight) through intraperitoneal injection. Three days post-phenylhydrazine treatment, the mice were infected with $2.0 \times 10^6$ parasites through tail vein injection. Peaks of gametocytemia usually appear 3 days post-infection. Male and female gametocytes were counted via Giemsa staining of thin blood smears. Gametocytemia was calculated as the ratio of male or female gametocytes over parasitized erythrocytes. All experiments were repeated three times independently.

### Exflagellation assay

Male gametocyte exflagellation was analyzed by counting *in vitro* formation of the exflagellation center. Briefly, 2.5 μl of mouse tail blood containing 4–6% gametocytemia was added to 100 μl exflagellation medium (RPMI 1640 supplemented with 10% fetal calf serum and 100 μM xanthurenic acid/XA) containing 1 μl of 200 unit/ml heparin and mixed thoroughly. After incubation at 22°C for 10 min, 10 μl of culture was transferred to the hemocytometer. Both the numbers of exflagellation centers and erythrocytes were counted under a 40× objective lens. The percentage of erythrocytes containing male gametocyte was counted from Giemsa-stained smears. The number of exflagellation centers per 100 male gametocytes was calculated as the exflagellation rate. All experiments were repeated three times.

### *In vitro* ookinete culture

*In vitro* culture for ookinete development was prepared as described previously (Gao *et al*, 2018). Mouse blood with 4–6% gametocytemia was collected in heparin tubes and immediately added to the ookinete culture medium (RPMI 1640 supplemented with 25 mM HEPES, 10% fetal calf serum, and 100 μM XA; pH 8.0). Parasites were cultured in the medium with a blood/medium volume ratio of 1:10 at 22°C. After 12 h of culture, the ookinetes were Giemsa-stained and analyzed for ookinete morphology. Ookinete conversion rate was calculated as the number of ookinetes

(both mature and immature) per 100 female gametocytes. The mature ookinete conversion rate was calculated as the number of mature ookinetes per 100 female gametocytes. All experiments were repeated three times.

## Mosquito transmission

Thirty female *A. stephensi* mosquitoes were allowed to feed on an anesthetized mouse carrying 4–6% gametocytemia for 30 min. For oocyst formation, mosquito midguts were dissected at day 7 or 8 post-blood-feeding and stained with 0.1% mercurochrome for oocyst counting. For salivary gland sporozoites counting, salivary glands from 20–30 mosquitoes were dissected at day 14 post-blood-feeding, and the number of sporozoites per mosquito was calculated. For sporozoite infection of mice, 10–15 infected mosquitoes were allowed to bite one anesthetized naïve mouse for 30 min.

## Parasite genetic cross

A similar amount ($1.0 \times 10^6$ parasites) of asexual blood-stage parasites from two different gene knockout strains were mixed to infect a phenylhydrazine-treated mouse through the tail vein injection. Three days after infection, the mouse with high gametocytemia was prepared for mosquito infection. At day 7 post-mosquito infection, mosquitoes were dissected for counting oocysts.

## Antibodies and antiserum

The primary antibodies used were as follows: rabbit anti-HA (Cell Signaling Technology, RRID:AB_1549585) (Western blot, 1:1,000 dilution; IFA, 1:1,000 dilution), rabbit anti-Myc (Cell Signaling Technology, RRID:AB_10692100) (Western blot, 1:1,000), mouse anti-Myc (Cell Signaling Technology, RRID:AB_331783) (IFA, 1:1,000), rabbit anti-GFP (Cell Signaling Technology, RRID:AB_1196615) (IFA, 1:1,000) mouse anti-α-Tubulin II (Sigma-Aldrich, RRID:AB_477583) (IFA, 1:1,000), rabbit anti-mCherry (Abcam, RRID:AB_2650480) (IFA, 1:1,000), and rabbit anti-IgG (Cell Signaling Technology, RRID:AB_1031062) (ChIP, 1:500). The secondary antibodies used were as follows: goat anti-rabbit IgG HRP-conjugated secondary antibody (Abcam, RRID:AB_955447) and goat anti-mouse IgG HRP-conjugated secondary antibody (Abcam, RRID:AB_955439) at 1:5,000 dilutions, the Alexa 555 goat anti-rabbit IgG secondary antibody (Thermo Fisher, RRID:AB_141784), Alexa 488 goat anti-rabbit IgG secondary antibody (Thermo Fisher, RRID:AB_10374301), Alexa 555 goat anti-mouse IgG secondary antibody (Thermo Fisher, RRID:AB_141822), and Alexa 488 goat anti-mouse IgG secondary antibody (Thermo Fisher, RRID:AB_2534069) at 1:1,000 dilutions. The antiserums used were prepared in our laboratory, including the rabbit anti-P28 (Western blot, 1:1,000; IFA, 1:1,000) and rabbit anti-BiP (Western blot, 1:1,000). Hoechst 33342 (Thermo Fisher, #62249) was used to visualize the nucleus.

## Immunofluorescence assay

Purified parasites were fixed using freshly prepared 4% paraformaldehyde (Sigma-Aldrich, P6148) in PBS for 15 min at room temperature and transferred to a 24-well cell plate containing a Poly-L-Lysine (Sangon Biotech, #E607015) pre-treated coverslip at the bottom. The fixed cells were then immobilized on the coverslip via centrifuging the plate at 650 *g* for 10 min and washed twice with PBS. The fixed cells were permeabilized with 0.1% Triton X-100 PBS solution for 7 min at room temperature, washed with PBS three times, blocked in 5% BSA solution for 60 min at room temperature, and incubated with the primary antibodies diluted in 3% BSA-PBS at 4°C for 12 h. The coverslip was incubated with fluorescent conjugated secondary antibodies for 1 h at room temperature and washed with PBS three times. Cells were stained with Hoechst 33342 (Thermo Fisher, #62249), mounted in 90% glycerol solution, and sealed with nail polish. All images were captured and processed using identical settings on a Zeiss LSM 780 laser scanning confocal microscopy.

## Protein extraction and Western blot

Protein extraction from asexual blood parasites, gametocytes, and ookinetes was performed using buffer A (50 mM Tris–HCl, pH 7.4, 150 mM NaCl, 1% NP-40, 0.5% sodium deoxycholate, 0.1% SDS) plus 1× complete protease inhibitor cocktail (MedChemExpress, #HY-K0010) and 1 mM PMSF (MedChemExpress, #HY-B0496). After ultrasonication, the protein solution was incubated on ice for 30 min before centrifugation at 12,000 *g* for 10 min at 4°C. The supernatant was applied to the Western blot analysis. Separate gels with different concentrations were prepared according to the molecular weight of the target proteins. Proteins were separated in SDS–PAGE and transferred to PVDF membrane (Millipore, #IPVH00010). The membrane was blocked in 5% skim milk TBST buffer for 60 min at room temperature and incubated with primary antibodies at 4°C overnight. After incubation, the membrane was washed three times with TBST and incubated with horseradish peroxidase-conjugated goat anti-rabbit or anti-mouse antibodies for 60 min at room temperature. The membrane was washed four times in TBST before enhanced chemiluminescence (Pierce, #32109) detection.

## Flow cytometry analysis and cell sorting

The infected mouse blood containing gametocytes was collected after induction by phenylhydrazine. After two washes with PBS, the cells were suspended in PBS with Hoechst 33342 (Thermo Fisher, #62249) for nuclei staining and applied for flow cytometry using BD LSR Fortessa flow cytometer (BD). Parasite-infected RBCs (iRBC) were first gated using the fluorescence signal of 405 nm (Hoechst 33342). GFP-positive male gametocytes and mCherry-positive female gametocytes were gated using 488 and 561 nm, respectively. For gametocyte sorting, the gametocytes of the *DFsc7* and *DFsc7*; Δ*ap2-o3* strains were enriched by centrifugation (1,900 × *g* for 10 min) on a 48% Nycodenz solution and then applied for cell sorting using MoFlo Astrios EQS (Beckman Coulter). Female gametocytes were sorted out through the signal of mCherry. After sorting, the purity of cells was assessed by FACS analysis. All the data were processed by FlowJo software.

## RNA preparation and quantitative RT–PCR

Total RNA of parasites was extracted using TRIzol reagent (Invitrogen, #15596026). Then, the total RNA was treated with DNase using

the Turbo DNA-free kit (Invitrogen, #AM1907). cDNA was synthesized using RevertAid reverse transcriptase (Thermo Fisher, #EP0441). For real-time quantitative PCR, the cDNA was diluted 1:20 with DEPC-treated water and quantified by SYBR Green PCR using iQ™ SYBR Green Supermix (Bio-Rad, #1708880) in the CFX96 Touch qPCR System (Bio-Rad). Gene-specific primers are listed in Appendix Table S2. Three biological replicates, with three technical replicates for each biological replicate, were performed for each tested gene. 18S rRNA was used as a reference gene for qRT–PCR. qRT–PCR data were analyzed using the ΔΔCt method.

## RNA-seq and data analysis

About $10^7$ of purified female gametocytes were used for total RNA extraction. After 2 μg of total RNA was extracted, mRNA was enriched by Oligo(dT) beads. The enriched mRNA was first fragmented using fragmentation buffer and then reverse transcribed to cDNA with random primers. Second-strand cDNA was synthesized by DNA polymerase I, RNase H, dNTP, and buffer. Then, the cDNA fragments were purified with QiaQuick PCR extraction kit, end-repaired, poly(A) added, and ligated with Illumina sequencing adapters. The ligation products were size-selected by agarose gel electrophoresis, PCR-amplified, and sequenced using Illumina HiSeqTM 2500 by Gene Denovo Biotechnology (Guangzhou, China). At least 45 million clean reads of sequencing depth were obtained for each sample. RNA-seq raw data were initially filtered by fastp (version 0.18.0) to obtain high-quality clean data. Short reads alignment tool Bowtie2 (version 2.2.8) was used for mapping reads to ribosome RNA (rRNA) database. The rRNA mapped reads will be removed. The remaining reads were further used in assembly and analysis of transcriptome. The rRNA removed reads of each sample were then mapped to reference genome by TopHat2 (version 2.0.3.12), respectively. Gene abundances were quantified by software RSEM. There were two steps for RSEM to quantify gene abundances. Firstly, a set of reference transcript sequences were generated and preprocessed according to known transcripts (in FASTA format) and gene annotation files (in GTF format). Secondly, RNA-seq reads were realigned to the reference transcripts by Bowtie alignment program and the resulting alignments were used to estimate gene abundances. To identify differentially expressed genes across samples or groups, the edgeR package (http://www.r-project.org/) was used. Trimmed mean of M-values normalization (TMM) method was used, and the gene expression level was normalized by using FPKM (Fragments Per Kilobase of transcript per Million mapped reads) method. We identified genes with a fold change ≥ 2 and a false discovery rate (FDR) < 0.05 in a comparison as significant DEGs. 1,625 genes were preliminarily identified as DEGs. To acquire more reliable DEGs, 348 members in *pir* and *fam* multigene families were excluded because of their various expression among different repeats.

## ChIP-qPCR

Mice infected with parasite were treated with sulfadiazine in drinking water for 24–32 h to kill asexual stage parasites. Blood with high gametocytemia was collected from mouse orbital sinus into heparin tubes and depleted of leukocytes using NWF Filter (ZhiXing Bio, China). Gametocytes were immediately fixed with 1% of methanol-free formaldehyde (Pierce, #28906) at room temperature for 10 min with gentle shaking for cross-linking DNA-proteins. Fixed cells were subjected to lysis in 0.84% $NH_4Cl$ for 10 min on ice and then washed twice with gametocyte maintenance buffer (GMB, containing 137 mM sodium chloride, 4 mM potassium chloride, 1 mM calcium chloride, 20 mM HEPES, 20 mM glucose, 4 mM sodium bicarbonate, 0.1% w/v bovine serum albumin, pH 7.25). The gametocytes were further enriched in a 60% Nycodenz gradient centrifugation and harvested for ChIP. ChIP assays were performed using SimpleChIP Plus Sonication Chromatin IP Kit (Cell Signaling Technology, #56383). The cross-linked cells were lysed and followed by chromatin-shearing using Covaris M220 focused ultrasonicator (Covaris, Inc) with the procedures: peak power 75W, 20% duty factor, 200 cycles per burst, and total treatment time of 600 s. The distribution and concentration of sheared chromatin were detected by agarose gel electrophoresis and NanoDrop 2000 (Thermo Fisher). 10 μg of chromatin was immunoprecipitated with anti-HA rabbit antibodies (Cell Signaling Technology, AB_1549585). As a control, the same amount of IgG antibody (Cell Signaling Technology, AB_1031062) was used. Immunoprecipitated chromatin collected with A + G magnetic beads was extensively washed and eluted with elution buffer. The input and ChIP samples were reverse cross-linked overnight at 65°C in the presence of Proteinase K (Thermo Fisher, #AM2546) and purified using the QIAquick Gel Extraction Kit (QIAGEN, #28704). The harvested DNA was subjected to qPCR analysis using the primers listed in Appendix Table S2.

## ChIP-seq

ChIP samples were quantified using a Qubit 2.0 Fluorometer (Invitrogen, USA) and qualified by Agilent Bioanalyzer 2100 (Agilent Technologies, USA). For each sample, at least 5 ng ChIP product was used for library preparation using VAHTS Universal Pro DNA Library Prep Kit (Vazyme, #ND-608). The ChIP product was treated with End Prep Enzyme Mix for end repairing, 5′ Phosphorylation, and dA-tailing in one reaction, followed by ligation to adaptors with a "T" base overhang. Adaptor-ligated DNA was recovered using AxyPrep Mag PCR Clean-up (Axygen, #MAG-PCR-CL-50) and amplified by PCR for 10 cycles using P5 and P7 primers, with both primers carrying sequences which can anneal with flowcell to perform bridge PCR and P7 primer carrying a six-base index allowing for multiplexing. PCR products were cleaned using AxyPrep Mag PCR Clean-up, validated using an Agilent 2100 Bioanalyzer, and quantified by Qubit 2.0 Fluorometer. Then, libraries with different indexes were multiplexed and loaded on an Illumina instrument (Illumina, USA). Sequencing was carried out using a 2 × 150 paired-end (PE) configuration; image analysis and base calling were conducted by the HiSeq Control Software (HCS) + OLB + GAPipeline-1.6 (Illumina) on the Illumina instrument by GENEWIZ (Suzhou, China).

## ChIP-seq data analysis

To remove technical sequences, including adapters, primers, fragments, and quality of bases lower than 20, pass filter data of fastq format were processed by Cutadapt (version 1.9.1) to obtain high-quality clean data. The reference genome sequences and gene

model annotation files of *P. yoelii* were downloaded from Plas-moDB 36. For mapping, the clean data were aligned with reference genome via software Bowtie. The mapping data (immunoprecipitated and input) were analyzed with the MACS2 (V2), using approximately $1.9 \times 10^7$ reads for IP and $4.3 \times 10^7$ reads for the input control in experiment 1, or $1.5 \times 10^7$ reads for IP and $3.1 \times 10^7$ reads for the input control in experiment 2. Dynamic Poisson distribution was used to calculate *P*-value of the specific region based on the unique mapped reads. The region is defined as a peak when *P*-value < 1e-5. Chromosome distribution, peak width, fold enrichment, significant level, and peak summit number were displayed. To predict the specific binding DNA motif of AP2-O3, the 300 bp of regions around summits of overlapped peaks from ChIP-seq experiment 2 were extracted and underwent MEME analysis (Bailey & Elkan, 1994). Control sequences are the similar length of region from the initiation codon of each peak-associated gene.

### Recombinant protein expression

A PCR product encoding 50 aa AP2 domain and 50 aa flanking coding region was inserted into the pET-GST vector. Recombinant proteins were expressed in *E. coli* BL21(DE3) strain (Invitrogen, #C600003). The transformed bacteria were inoculated in LB medium supplemented with 100 μg/ml of ampicillin (Sangon Biotech, #A100339) and incubated overnight at 37°C with shaking (200 rpm). Next, an inoculum (1:50) was made in a fresh medium and culture was incubated at 37°C with shaking (200 rpm). Until the culture reached an optical density (OD600) of 0.4 to 0.6, isopropyl-β-d-thiogalactoside (Sangon Biotech, #A600168) was added to a final concentration of 0.2 mM. The culture was induced at 16°C overnight and affinity purified using glutathione resin (Thermo Fisher, #16100) according to the manufacturer's instructions. The efficiency of protein purification was estimated by SDS–PAGE and Coomassie blue staining.

### EMSA

DNA binding of purified N-terminal GST fusions of AP2 domains of AP2-O3 with DNA probe sequences was analyzed by EMSA. Single-stranded oligonucleotides containing the recognition motif and their corresponding complementary oligonucleotides were synthesized and purchased from GENEWIZ (CHINA) as labeled (59-biotinylated and HPLC purified) and unlabeled sequences. Complementary single-stranded oligonucleotides were annealed to create double-stranded probes and used for EMSA as labeled and unlabeled target probes for the DBD of AP2-O3. The EMSA was conducted using the LightShift chemiluminescence EMSA kit (Thermo Fisher, #20148) following the instructions of the manufacturer. In brief, the purified GST fusion of AP2-O3 was pre-incubated with the labeled probe the binding reaction containing binding buffer, 1 μg poly(dI-dC), 50% glycerol, 100 mM MgCl$_2$, and 1% NP40 at room temperature (22°C) for 10 min. The unlabeled probe (200-fold excess to the labeled probe) was then added as a competitor, and the reaction was incubated for further 20 min at room temperature. The reaction was fractionated using 6% PAGE and transferred to a nylon membrane (Ambion, #10104) according to the manufacturer's instructions. Specific binding of the AP2 domain with the target motif was detected as an upward shift using the Chemiluminescence Nucleic Acid Detection Module (Thermo Fisher, #89880), as per the manufacturer's instructions.

### Bioinformatics analysis and tools

The genomic sequences of target genes were downloaded from the PlasmoDB database (http://plasmodb.org/plasmo/). The sgRNAs of a target gene were designed using EuPaGDT (http://grna.ctegd. uga.edu/). Amino acid sequence alignment was performed using MEGA5.0. Flow cytometry data were analyzed using FlowJo v10. The MEME Suite was used to discover enriched motifs in the DNA sequence. Gene ontology (GO) analysis was performed with the OmicShare tools (http://www.omicshare.com/tools), with GO terms taken from GO consortium (http://geneontology.org/page/download-annotations), with predicted GO terms downloaded from the PlasmoDB database, and with terms collected by manual categorizing. GO term enrichments was analyzed relative to the background of all genes. A term was identified as significant if the *P*-value was below 0.05.

### Quantification and statistical analysis

Statistical analysis was performed using GraphPad Software 8.0. Two-tailed Student's *t*-test or Mann–Whitney test was used to compare differences between treated groups. *P*-value in each statistical analysis was indicated within the figures.

## Data availability

The datasets produced in this study are available in the following databases:

- RNA-seq data: Gene Expression Omnibus GSE157456 (https://www.ncbi.nlm.nih.gov/geo/query/acc.cgi?acc=GSE157456)
- ChIP-seq data: Gene Expression Omnibus GSE157454 (https://www.ncbi.nlm.nih.gov/geo/query/acc.cgi?acc=GSE157454)

**Expanded View** for this article is available online.

### Acknowledgements

We thank Dr. Bo Wang, Dr. Mathieu Brochet, and Dr. Rita Tewari for the comments on this manuscript. This work was supported by the National Natural Science Foundation of China (31772443, 31872214, 31970387, and 32000445), the Natural Science Foundation of Fujian Province (2019J05010), and the 111 Project sponsored by the State Bureau of Foreign Experts and Ministry of Education of China (BP2018017).

### Author contributions

ZKL, JPG, and CL generated the modified parasites. ZKL and JPG conducted the phenotype analysis, IFA assay, image analysis, mosquito experiments, and biochemical experiments. ZGY performed the bioinformatics analysis. ZKL, HTC, and JY analyzed the data. HTC and JY supervised the work. JY wrote the manuscript.

### Conflict of interest

The authors declare that they have no conflict of interest.

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
