## [Review Process File · EMBO Reports]

Plasmodium transcription repressor AP2-O3 regulates sex-specific identity of gene expression in female gametocytes

Zhenkui Li, Huiting Cui, Jiepeng Guan, Cong Liu, Zhengang Yang, and Jing Yuan
DOI: [10.15252/embr.202051660](https://doi.org/10.15252/embr.202051660)

Corresponding author(s): Jing Yuan (yuanjing@xmu.edu.cn)

Review Timeline:

Submission Date:	2nd Sep 20
Editorial Decision:	29th Oct 20
Revision Received:	2nd Dec 20
Editorial Decision:	26th Jan 21
Revision Received:	27th Jan 21
Accepted:	5th Feb 21

Editor: Deniz Senyilmaz Tiebe

Transaction Report:

Dear Prof. Yuan,

Thank you for submitting your manuscript to EMBO Reports.

I apologize for this unusual delay in getting back to you, it took longer than anticipated to receive the referee reports. Three referees agreed to review your manuscript. So far, we have received two referee reports that are copied below. Given that both referees are in fair agreement that you should be given a chance to revise the manuscript, I would like to ask you to begin revising your study along the lines suggested by the referees.

Please note that this is a preliminary decision made in the interest of time, and that it is subject to change should the third referee offer very strong and convincing reasons for this. As soon as/if we receive the final report on your manuscript, we will forward it to you as well.

Referees express interest in the proposed role of AP2-O3 in regulation of sex-specific identity of gene expression in female gametocytes. However, they also raise important concerns that need to be addressed to consider publication here.

I find the reports informed and constructive, and believe that addressing the concerns raised will significantly strengthen the manuscript. As the reports are below, and I think all points need to be addressed, I will not detail them here.

Given these constructive comments, we would like to invite you to revise your manuscript with the understanding that the referee concerns (as in their reports) must be fully addressed and their suggestions taken on board. Please address all referee concerns in a complete point-by-point response. Acceptance of the manuscript will depend on a positive outcome of a second round of review. It is EMBO reports policy to allow a single round of revision only and acceptance or rejection of the manuscript will therefore depend on the completeness of your responses included in the next, final version of the manuscript.

*** Temporary update to EMBO Press scooping protection policy:

We are aware that many laboratories cannot function at full efficiency during the current COVID-19/SARS-CoV-2 pandemic and have therefore extended our 'scooping protection policy' to cover the period required for a full revision to address the experimental issues highlighted in the editorial decision letter. Please contact the scientific editor handling your manuscript to discuss a revision plan should you need additional time, and also if you see a paper with related content published elsewhere.***

IMPORTANT NOTE: we perform an initial quality control of all revised manuscripts before re-review. Your manuscript will FAIL this control and the handling will be DELAYED if the following APPLIES:
1. A data availability section providing access to data deposited in public databases is missing

(where applicable).

2. Your manuscript contains statistics and error bars based on $n=2$. Please use scatter plots in these cases.

Supplementary/additional data: The Expanded View format, which will be displayed in the main HTML of the paper in a collapsible format, has replaced the Supplementary information. You can submit up to 5 images as Expanded View. Please follow the nomenclature Figure EV1, Figure EV2 etc. The figure legend for these should be included in the main manuscript document file in a section called Expanded View Figure Legends after the main Figure Legends section. Additional Supplementary material should be supplied as a single pdf labeled Appendix. The Appendix includes a table of content on the first page with page numbers, all figures and their legends. Please follow the nomenclature Appendix Figure Sx throughout the text and also label the figures according to this nomenclature. For more details please refer to our guide to authors.

Please note that for all articles published beginning 1 July 2020, the EMBO Reports reference style will change to the Harvard style for all article types. Details and examples are provided at <https://www.embopress.org/page/journal/14693178/authorguide#referencesformat>

2) individual production quality figure files as .eps, .tif, .jpg (one file per figure).

3) a .docx formatted letter INCLUDING the reviewers' reports and your detailed point-by-point responses to their comments. As part of the EMBO Press transparent editorial process, the point-by-point response is part of the Review Process File (RPF), which will be published alongside your paper. For more details on our Transparent Editorial Process, please visit our website:

<https://www.embopress.org/page/journal/14693178/authorguide#transparentprocess>

4) a complete author checklist, which you can download from our author guidelines (<http://embor.embopress.org/authorguide>). Please insert information in the checklist that is also reflected in the manuscript. The completed author checklist will also be part of the RPF.

5) Please note that all corresponding authors are required to supply an ORCID ID for their name upon submission of a revised manuscript (<https://orcid.org/>). Please find instructions on how to link your ORCID ID to your account in our manuscript tracking system in our Author guidelines (<http://embor.embopress.org/authorguide>).

6) We replaced Supplementary Information with Expanded View (EV) Figures and Tables that are collapsible/expandable online. A maximum of 5 EV Figures can be typeset. EV Figures should be cited as 'Figure EV1, Figure EV2' etc... in the text and their respective legends should be included in the main text after the legends of regular figures.

- For the figures that you do NOT wish to display as Expanded View figures, they should be bundled together with their legends in a single PDF file called *Appendix*, which should start with a short Table of Content. Appendix figures should be referred to in the main text as: "Appendix Figure S1, Appendix Figure S2" etc. See detailed instructions regarding expanded view here: <<http://embor.embopress.org/authorguide#expandedview>>.

7) We would also encourage you to include the source data for figure panels that show essential data.

Numerical data should be provided as individual .xls or .csv files (including a tab describing the data). For blots or microscopy, uncropped images should be submitted (using a zip archive if multiple images need to be supplied for one panel). Additional information on source data and instruction on how to label the files are available <<http://embor.embopress.org/authorguide#sourcedata>>.

8) Our journal encourages inclusion of *data citations in the reference list* to directly cite datasets that were re-used and obtained from public databases. Data citations in the article text are distinct from normal bibliographical citations and should directly link to the database records from which the data can be accessed. In the main text, data citations are formatted as follows: "Data ref: Smith et al, 2001" or "Data ref: NCBI Sequence Read Archive PRJNA342805, 2017". In the Reference list, data citations must be labeled with "[DATASET]". A data reference must provide the database name, accession number/identifiers and a resolvable link to the landing page from which the data can be accessed at the end of the reference. Further instructions are available at <<http://embor.embopress.org/authorguide#datacitation>>.

9) Please make sure to include a Data Availability Section before submitting your revision - if it is not applicable, make a statement that no data were deposited in a public database. Primary datasets (and computer code, where appropriate) produced in this study need to be deposited in an appropriate public database (see <<http://embor.embopress.org/authorguide#dataavailability>>).

The accession numbers and database should be listed in a formal "Data Availability " section (placed after Materials & Method) that follows the model below. Please note that the Data Availability Section is restricted to new primary data that are part of this study.

Data availability

10) Regarding data quantification, please ensure to specify the name of the statistical test used to generate error bars and P values, the number (n) of independent experiments underlying each data point (not replicate measures of one sample), and the test used to calculate p-values in each figure legend. Discussion of statistical methodology can be reported in the materials and methods section, but figure legends should contain a basic description of n, P and the test applied.

Please note that error bars and statistical comparisons may only be applied to data obtained from at least three independent biological replicates.

I look forward to seeing a revised version of your manuscript when it is ready. Please let me know if you have questions or comments regarding the revision.

Yours sincerely,

Deniz Senyilmaz Tiebe

Deniz Senyilmaz Tiebe, PhD
Editor
EMBO Reports

Referee #1:

Evaluation

Li et al Plasmodium transcription repressor AP2-O3 regulates sex-1 specific identity of gene expression in female gametocytes

They show that expression of the AP2-O3 (O3) is necessary to repress male gametocyte transcription in the female lineage. It does not prevent the expression of female genes however, highly expressed female genes tend to have lower gene expression in the absence of O3. The extent of commitment to the production of male gametocytes is unaffected. The authors build a picture of a protein that is briefly expressed in female gametocyte lineage to preserve the integrity of the female transcriptome through prevention of male gene expression. Furthermore, the absence of O3 leads to a cell that can be fertilized but fails to develop into a mature functional ookinete which was the original recorded phenotype for this gene. The timing of expression is shown to be important as overexpression in zygotes again produces a non-infective ookinete. Analysis of the O3-repressed genes helps identify a motif upstream of the target genes that a recombinant DNA binding domain of O3 is able to bind. Female gene expression is also altered (abundant genes down, lower abundance genes up) but this is merely observational and no further analyses are conducted to provide any mechanism.

Major comments.

The English is idiosyncratic and will require substantial revision/editing to make the concepts and data clear to a general reader

Much of what is reported in the first stages of the manuscript has been published by the authors before (Zhang et al. 2017), so could either be moved to the supplementary data or dropped and referred to in the published literature.

- a. "AP2-O3 is expressed in the female gametocytes and mature oocyst"-that was already shown (fig3 of this manuscript) using the same techniques (IFA) as employed here.
- b. "AP2-O3 is essential for ookinete formation and mosquito transmission" - again shown in Zhang et al. 2017. Using the same set of assays/knockout lines.
- c. "AP2-O3 null female gametocytes fail to develop mature fertile gametes" - partially as the effect of the ap2o3 KO on P28 expression was published as well by the authors.

The RNA-seq analysis of ap2-o3 gametocytes and developing ookinetes have been performed previously in *P. berghei*, a sister species very closely related to *P. yoelii* (Modrzynska et al). It would be reasonable to expect to see the comparison of the two datasets.

There is something very strange about the data. The authors make frequent use of a double reporter line and perform the AP2-O3 ko in this background. Subsequent analysis of purified AP2-O3 ko female gametocytes reveals that the gene PY17X_0418900 (Fig S5 gene 19) that provides the promoter to drive expression of GFP in the male lineage is also upregulated. Why is this not reflected in the expression of GFP in this mutant? Ccp2 is a very late onset gene (later than PY17X_0418900 at least in *P. berghei*) and so any RFP positive AP2-O3ko female cell should be mature and expressing GFP at least to some extent yet the FACS traces in S4F don't give any hint of such an event. The wt line appears to be double positive to some extent (noted also below). Some of the significant changes in the transcriptome could be very easily explained by the contamination of the mutant prep with male gametocytes if the expression of GFP/mcherry is missorted.

The overall extent of the genetic dysregulation (authors cite ~1400 differentially expressed genes which is >25% of the parasite transcriptome) makes it difficult to distinguish which changes are due directly to ap2-o3 and which are secondary. There is no so much overlap between the genes identified by RNA-seq and ChIP-seq which suggests that most of the observed changes could be secondary. Perhaps the data could be analyzed to search for additional regulators which can explain downstream effects? For example, perform a motif search in the promoter of the genes which are dysregulated but don't have ChIP peaks associated with them. Does the AP2-O3 motif appear enriched in the downregulated or upregulated female genes in the O3ko background? If, as it appears, the DNA motif was identified by using all peaks in the genome. As ap2o3 seems to fulfil both repressive and activating function, perhaps it would be good to analyze separately the peaks associated with up- and down-regulated genes in case some other cryptic motif is present

Is the total mRNA content of the cell altered - are some of these changes in the female transcriptome nothing more than a reflection of the capacity of the cell to transcribe mRNA?

The authors have a habit of reporting work in *Toxoplasma* as if it were work carried out in *Plasmodium* e.g. refs 25, 35 & 37 (there may be others, the search was not exhaustive) - this is misleading to the reader and must be corrected.

L138. The conclusion is that AP2-O3 has a conserved function, surely

Figure S4F there appears to be a population of double positive cells expressing both "gender specific" markers in the wt and, although fewer cells are analyzed, to a lesser extent in the O3 ko. Does this confuse the subsequent tagging analysis of male specific genes in the female background of the AP2-O3 ko?

Although the male gametocytes appear to be normal beyond their gross functional characterization at the level of their ability to generate ookinetes with wt female gametes. Transcriptome would be interesting.

Figure 5 - all the male-specific genes chosen for demonstration of expression in "female" gametocytes of the AP2-O3ko background are nuclear - is this exclusively the case? It would appear not to be so are the cytoplasmic protein encoding transcripts not translated? It is reported that male genes are not as abundant as transcripts in O3ko females but have the most abundant male transcripts from these females been tested for expression? It would be useful to have proceeded down the list of abundance rather than just picking some genes which in the absence of a rationale seems to have been what happened.

The model presented in Figure S9 doesn't really summarize the data. In the absence of O3 then there is a superabundance of male transcripts as well as a change in the stoichiometry of the constituency of the female transcriptome. Failure to generate a functional infective ookinete could well be due to male protein subversion of female protein function as the changes to the female transcriptome per se. The model and surrounding discussion should be reworked to more accurately reflect all the different possible speculations.

Minor comments

General issues with the use of the definite and indefinite pronouns and syntax which can be sorted with editing.

ALL the references should be checked as many are not listed correctly: volume and page numbers often seem to be missing

L 13: "distinct repertoireS"

L72: "dogma" is not the correct term

L134. The sentence beginning on that line does not make sense - pop a verb in there.

L153 Ref 24 does not speak to translational repression of P28 - please find an appropriate reference

L155: please quantitate the reduced abundance of P28 as it does not appear to be markedly once control loading is taken into account.

L158: There is no mention of the word zygote in the cited reference!! 25 Frenal et al. Use the literature accurately!!! What is the correct reference!

L206 - sentence beginning here needs attention.

Fig 1A Can the authors comment on the apparent expression of AP2-O3 in oocyst sporozoites? This seems to have been ignored in the description of the results.

Fig 1B, the scale bars on this CANNOT be uniformly 5um

1C. spelling of ookinete

Fig 2E-H labels on the x axis?

FicS4G - Identify the samples clearly - this is just an analysis of purified female gametocytes, correct?

Referee #2:

The manuscript by Li et al entitled 'Plasmodium transcription repressor AP2-O3 regulates sex-specific identity of gene expression in female gametocytes' reports a functional study of transcription factor AP2-O3, which belongs to the repressor gene ApiAP2 Transcription Factors family in Plasmodium. The study convincingly shows that AP2-O3 is specifically localized in female gametocytes by IFA studies using HA epitope tagged AP2-O3 in *P. yoelli* parasites, and the transcriptional role for AP2-O3 is revealed by immunoblotting and in vitro exflagellation assays using AP2-O3-deficient *P. yoelli* parasites. This revealed that AP2-O3 is essential for ookinete formation and that male gametocytes deficient of AP2-O3 fail to develop into fertile male gametocytes.

In the second part of the study large scale expression studies are reported. RNA-Seq analysis is performed for gaining insights into changes in transcript levels in female gametocytes in AP2-O3 KO parasites compared to wild type female gametocytes. ChIP-seq analysis identified an eight-base DNA binding sequence as promoter sequence for specific male genes.

I do have a few points of concern for the second part of the study for the authors to be addressed:

1) Paragraph 'Transcriptome analysis of female gametocytes' (lines175-200).

I find this paragraph slightly confusing in the way transcriptome analysis results are presented. It is rather difficult to understand which comparative data analysis is presented. The study has acquired transcriptome data for 'wild type' DFsc7 male and female *P.yoelli* parasites, and mutant DFsc7; Δ ap2-o3 male and female *P.yoelli* parasites, but does not present expression results for all the acquired data. I think it would be easier to understand when all data is presented first, and how many statistically relevant upregulated transcripts are determined in female WT DFsc7parasites compared to male WT DFsc7parasites and vice versa? The same for male and female DFsc7; Δ ap2-o3 parasites.

The study only reports the identification of 1141 up-regulated genes and 136 down-regulated genes (> 2-fold) in the female gametocytes of DFsc7; Δ ap2-o3 compared to WT DFsc7 parasites.

How do these differentially expressed transcripts in DFsc7; Δ ap2-o3 parasites compare to expression in WT male parasites. Are the 1141 up-regulated genes in DFsc7; Δ ap2-o3 female gametocytes expressed in higher levels in WT male gametocytes compared to WT female gametocytes? This would demonstrate a male specific function as reported in paragraph 'AP2-O3 represses male gene transcription in female gametocytes' (line 203-217) that these up-regulated genes are involved in male specific processes such as DNA replication, so I would expect these genes to be upregulated in WT male gametocytes relative to WT female gametocytes. This can be verified from the acquired transcriptome data.

I think it would help when expression data is provided as supplemental tables (excel or txt) with DEGs indicated.

Regarding the RNA Seq data processing procedure, I couldn't find much detail in the RNA seq Material and Methods section (652-661) about data processing steps, noise level threshold, and data normalisation procedure for female and male gametocyte samples. How many parasites were sorted and how much RNA was extracted from purified male and female parasites? Was a data normalisation procedure applied to RNA seq data from purified male and female parasites? How many transcripts were identified in male and female gametocytes before and after applying a noise level threshold for the RNA seq data?

Fig S4 shows a cross correlation heatmap of global gene expression between samples of two strains DFsc7 WT and DFsc7; Δ ap2-o3 gametocytes. I understand that a perfect correlation of 1.00 is determined for samples with itself (eg DFsc7 replicate 1 with DFsc7 replicate 1) but it is very unlikely that perfect correlations are observed between replicate samples (eg DFsc7 replicate 1 with DFsc7 replicate 2), and I am wondering if reported correlation coefficients (1.00) for DFsc7 are correct.

2. Paragraph 'Decreased expression of highly expressed female-specific genes in the AP2-O3 null female gametocytes' (lines 297 - 341).

This paragraph reports differences in gene expression and gene expression profiles supported by figure 7. Gene expression levels are represented as a heatmap with genes sorted by abundance levels in FPKM.

Small point: Which value represent the FPKM signal at left columns from heatmap figures, sum of all samples, or average of all samples, any normalisation applied between samples?

Expression patterns between samples are difficult to observe in presented heatmaps, and I think clustered heatmaps would provide a much better visual representation of expression profiles between the two parasite strains than the heatmaps in figure 7.

Observations are made for differences in expression patterns between abundant transcripts (FPKM>1500) and low abundant transcripts (FPKM<1500). How much of these differences is due to expression level variation in the low abundant transcripts? Are there any DEGs in the low abundant transcript group?

The authors observe small RNAseq expression differences for DOZI and CITH genes between wild type and ap2-o3 deficient parasites, while qRT-PCR, IFA and western blotting experiments reveal no significant expression differences at gene and protein level. This indicates to me that the small RNAseq expression differences for DOZI and CITH are not statistically significant and that a statistical analysis of all RNA seq data should be included.

3. Paragraph 'Sequential expression of AP2-O3 and AP2-O before and after female gametogenesis' (lines 376-400)

This paragraph presents some functional result for AP2-O, which is another member of the

ApiAP2 Transcription Factors. However, I don't see the relevance for these experiments in this study, and reported results may not be of great importance as these are not part of the Discussion.

Response to Reviewer Comments:**Referee #1:**

Li et al Plasmodium transcription repressor AP2-O3 regulates sex-1 specific identity of gene expression in female gametocytes.

They show that expression of the AP2-O3 (O3) is necessary to repress male gametocyte transcription in the female lineage. It does not prevent the expression of female genes however, highly expressed female genes tend to have lower gene expression in the absence of O3. The extent of commitment to the production of male gametocytes is unaffected. The authors build a picture of a protein that is briefly expressed in female gametocyte lineage to preserve the integrity of the female transcriptome through prevention of male gene expression. Furthermore, the absence of O3 leads to a cell that can be fertilized but fails to develop into a mature functional ookinete which was the original recorded phenotype for this gene. The timing of expression is shown to be important as overexpression in zygotes again produces a non-infective ookinete. Analysis of the O3-repressed genes helps identify a motif upstream of the target genes that a recombinant DNA binding domain of O3 is able to bind. Female gene expression is also altered (abundant genes down, lower abundance genes up) but this is merely observational and no further analyses are conducted to provide any mechanism.

Major comments.

The English is idiosyncratic and will require substantial revision/editing to make the concepts and data clear to a general reader.

Response:

We had the manuscript edited by a professional person. The changes in the text are highlighted with red letters.

Much of what is reported in the first stages of the manuscript has been published by the authors before (Zhang et al. 2017), so could either be moved to the supplementary data or dropped and referred to in the published literature.

a. "AP2-O3 is expressed in the female gametocytes and mature oocyst"-that was already shown (fig3 of this manuscript) using the same techniques (IFA) as employed here.

Response:

In this work, we provided evidences supporting female gametocyte-specific expression of AP2-O3 from four independent tagged strains (*ap2-o3::6HA*, *6HA::ap2-o3*, *ap2-o3::mScarlet*, and *ccp2::mCherry;ap2-o3::6HA*). In our previous work (Zhang *et al.* 2017, PMID: 29233900), only one strain (*ap2-o3::6HA*) was generated and analyzed. But we agreed with the reviewer's suggestion, and moved the Figure 1 to Figure EV1.

b. "AP2-O3 is essential for ookinete formation and mosquito transmission" - again shown in Zhang *et al.* 2017. Using the same set of assays/knockout lines.

Response:

In our 2017 work (Zhang *et al.* 2017, PMID: 29233900), only the *ap2-o3* knockout mutant was generated and analyzed. In this manuscript, we performed genetic complementation in the *ap2-o3* knockout mutant and confirmed that the transmission block defect is indeed caused by the *ap2-o3* knockout. Furthermore, we knocked out the *ap2-o3* in the *ap2-o3::6HA* strain background and observed the similar defect. In addition, we showed evidences that AP2-O3 from human malaria parasite *P. falciparum* is functionally conserved. We believed this figure provides critical information supporting causative role of AP2-O3 in ookinete formation and mosquito transmission of malaria parasite, and decided to keep this part as the revised Figure 1.

c. "AP2-O3 null female gametocytes fail to develop mature fertile gametes" - partially as the effect of the *ap2o3* KO on P28 expression was published as well by the authors. The RNA-seq analysis of *ap2-o3* gametocytes and developing ookinetes have been performed previously in *P. berghei*, a sister species very closely related to *P. yoelii* (Modrzynska *et al.*). It would be reasonable to expect to see the comparison of the two datasets.

Response:

Thanks for reviewer's suggestion. As reviewer mentioned, Modrzynska *et al* indeed performed the transcriptome of the *ap2-o3* gene-disrupted gametocytes in *P. berghei* (Modrzynska *et al.* 2017, PMID: 28081440). However, they collected and analyzed the gametocyte mixtures containing both male and female lineages, but not female gametocytes according to their description in Material and Method (PMID: 28081440). As we revealed that the female-specific AP2-O3 plays a transcription repressor role for male genes in the female gametocytes, it is important and necessary to purify the female gametocytes for comparison of female gametocyte's transcriptome between wt and *ap2-o3* disrupted parasites. In Modrzynska *et al* study, they did not report the upregulation of male genes, which could be likely explained by the gametocyte mixtures used. We believed it is not reasonable to compare these two datasets between theirs (male and female gametocyte mixtures) and ours (purified female gametocytes).

There is something very strange about the data. The authors make frequent use of a double reporter line and perform the AP2-O3 ko in this background. Subsequent analysis of purified AP2-O3 ko female gametocytes reveals that the gene PY17X_0418900 (Fig S5 gene 19) that provides the promoter to drive expression of GFP in the male lineage is also upregulated. Why is this not reflected in the expression of GFP in this mutant? Ccp2 is a very late onset gene (later than PY17X_0418900 at least in *P. berghei*) and so any RFP positive AP2-O3ko female cell should be mature and expressing GFP at least to some extent yet the FACS traces in S4F don't give any hint of such an event. The wt line appears to be double positive to some extent (noted also below). Some of the significant changes in the transcriptome could be very easily explained by the contamination of the mutant prep with male gametocytes if the expression of GFP/mcherry is missorted.

Response:

Because AP2-O3 is restrictedly expressed in female gametocytes but not male gametocytes (revised Figure EV1), and *ap2-o3* disrupted parasites developed normal male gametocytes and male gametes (revised Figure 2A and 2F). These results indicted no effect of AP2-O3 disruption on male gametocytes and male gamete

formation. Therefore, in this study, only female gametocytes (but not male gametocytes) were sorted out by FACS based on a double fluorescent reporter parasite line *DFsc7* we developed (Liu *et al.* 2018, PMID: 30040976). In this line, the genes encoding GFP and mCherry were C-terminally knocked into the downstream of endogenous genes *dhc1* (PY17X_0418900) and *ccp2*, respectively. Endogenous DHC1::GFP is restrictedly expressed in male gametocytes while Ccp2::mCherry expressed in female gametocytes.

We agreed with the reviewer's comments about the male gene *dhc1* (PY17X_0418900, gene No.19). As the reviewer noticed, this gene *dhc1* didn't show much difference in female gametocytes between *DFsc7* and *DFsc7;Δap2-o3* parasites at both mRNA (gene No.19 in revised Figure EV3) and protein levels (revised Figure EV2F). However, we can observe clear increase in mRNA level of other >30 male genes displayed (revised Figure EV3). So we could delete three genes (No.19: PY17X_0418900, No.20: PY17X_0508400, and No.21: PY17X_0927400) in the gene list (revised Figure EV3) if it is necessary.

For the double positive cells in FACS analysis of the revised Figure EV2F, we explained the reasons in the previous paper (Liu *et al.* 2018, PMID: 30040976). Below are the results from the Fig S1 of that paper (PMID: 30040976). In FACS analysis, the double-positive cell population appeared in 17XNL-infected mice only pretreated with phenylhydrazine (PHZ), which is commonly used for inducing gametocytogenesis (gametocyte formation) of rodent malaria parasites. The fluorescent microscopy further revealed that the double-positive cell populations are some subset (unidentified) of uninfected RBC, but not parasite-infected RBC. In our

practice, the *P. yoelii* DFsc7 line is widely used for purifying sex-specific gametocytes via FACS sorting.

The overall extent of the genetic dysregulation (authors cite ~1400 differentially expressed genes which is >25% of the parasite transcriptome) makes it difficult to distinguish which changes are due directly to ap2-o3 and which are secondary. There is no so much overlap between the genes identified by RNA-seq and CHiP-seq which suggests that most of the observed changes could be secondary. Perhaps the data could be analyzed to search for additional regulators which can explain downstream effects? For example, perform a motif search in the promoter of the genes which are

dysregulated but don't have ChiP peaks associated with them. Does the AP2-O3 motif appear enriched in the downregulated or upregulated female genes in the O3ko background? If, as it appears, the DNA motif was identified by using all peaks in the genome. As ap2o3 seems to fulfil both repressive and activating function, perhaps it would be good to analyze separately the peaks associated with up- and down-regulated genes in case some other cryptic motif is present.

Q1: Perhaps the data could be analyzed to search for additional regulators which can explain downstream effects? For example, perform a motif search in the promoter of the genes which are dysregulated but don't have ChiP peaks associated with them.

Response:

According to reviewer's suggestion, we performed a motif search in the promoters (1.5 kb sequences upstream of the "ATG" starting codon) of the 1141 up-regulated genes. Below are the top 3 motifs detected using the MEME analysis, and the No.2 hit shows some similarity with the eight-base DNA motif of "T(G/A)T(G/A)TGCA" identified in this study.

	Logo	E-value ?	Sites ?	Width ?	More ?	Submit/Download ?
1.		4.8e-010	482	6	↓	→
2.		1.6e-010	985	8	↓	→
3.		1.8e+000	854	8	↓	→

We also performed a motif search in the promoters (1.5 kb sequences upstream of the "ATG" starting codon) of the 136 down-regulated genes. Below are the top 3 motifs detected using the MEME analysis, but no hit shows similarity with the eight-base DNA motif of "T(G/A)T(G/A)TGCA" identified in this study.

	Logo	E-value ?	Sites ?	Width ?	More ?	Submit/Download ?
1.		1.0e-018	116	8	↓	→
2.		4.9e+002	33	8	↓	→
3.		2.0e+005	39	8	↓	→

Q2: Does the AP2-O3 motif appear enriched in the downregulated or upregulated female genes in the O3ko background? If, as it appears, the DNA motif was identified by using all peaks in the genome. As ap2o3 seems to fulfil both repressive and activating function, perhaps it would be good to analyze separately the peaks associated with up- and down-regulated genes in case some other cryptic motif is present.

Response:

The AP2-O3 motif “T(G/A)T(G/A)TGCA” are enriched in the promoter of the up-regulated genes but not the down-regulated genes. These results are agreement with our speculation that AP2-O3 mainly plays a transcription repressive function in the female gametocytes.

Is the total mRNA content of the cell altered - are some of these changes in the female transcriptome nothing more than a reflection of the capacity of the cell to transcribe mRNA?

Response:

First, in this study, three biological replicates in each group of WT and KO parasites were collected for RNA-seq. The RNA-seq results indicated that more than 45 million clean reads of sequencing depth were obtained in each sample, which is far more than the required transcriptome coverage. The numbers of transcribed genes detected in each sample are comparable: wt-1:5269 ; wt-2:5250 ; wt-3:5240 ; ko-1:5294 ; ko-2:5313 ; ko-3:5350.

Second, since that FPKM (fragments per kilobase of exon model per million reads mapped) is a normalized estimation of gene expression in RNA-seq. We calculated the sum of the FPKM value for all genes detected in WT and KO parasites, and found no marked difference between WT and KO parasites (844989±21016 in WT, 806015±16620 in the KO parasites).

Third, if the AP2-O3 disruption results in the defect of mRNA transcription in female gametocytes, it should be observed that downregulation of transcription occurs in all the genes, but not certain sets of genes.

Together, these analyses don't suggest altered capacity of mRNA transcription in the absence of AP2-O3.

The authors have a habit of reporting work in *Toxoplasma* as if it were work carried out in *Plasmodium* e.g. refs 25, 35 & 37 (there may be others, the search was not exhaustive) - this is misleading to the reader and must be corrected.

Response:

We are sorry for our incorrect citation. We have changed the “On the other hand, *Plasmodium* ApiAP2 TFs are found to associate with chromatin modifying enzymes, such as histone deacetylase³⁵ and histone acetyltransferase^{36, 37}.” to “ On the other hand, ApiAP2 TFs are found to associate with transcription regulators or chromatin remodelers such as BDP1, BDP2, and histone acetyltransferase in the *Plasmodium* (Josling *et al*, 2015; LaCount *et al*, 2005; Santos *et al.*, 2017; Toenhake *et al*, 2018), and histone deacetylase and acetyltransferase in the *Toxoplasma. gondii* (Dixon *et al*, 2010; Saksouk *et al*, 2005) ”.

L138. The conclusion is that AP2-O3 has a conserved function, surely

Response:

We changed “These results indicate that AP2-O3 is essential for ookinete formation and mosquito transmission” to “Together, these results indicate that AP2-O3 has a conserved role in promoting ookinete formation and mosquito transmission”.

Figure S4F there appears to be a population of double positive cells expressing both "gender specific" markers in the wt and, although fewer cells are analyzed, to a lesser extent in the O3 ko. Does this confuse the subsequent tagging analysis of male specific genes in the female background of the AP2-O3 ko?

Response:

In the above, we have responded to the reviewer's same comments about the double positive cells in FACS analysis of the revised Figure EV2F. The double-positive cell populations are some subset of uninfected RBC, but not parasite-infected RBC in mice pretreated with phenylhydrazine (PHZ) for inducing gametocytes. In our

practice of inducing gametocytes in rodent malaria parasite infected-mice via PHZ pretreatment, it is quite often that variable level (0.5 to 5%) of female and male gametocytes (gametocytemia) are produced within the infected mice. This is the reason for the variable numbers of male gametocyte population before sorting observed in the revised Figure EV2F. Importantly, in this study, only mCherry⁺ gametocytes (female) were sorted out for transcriptome analysis. After sorting, highly purified female gametocytes were obtained (99% and 99.2% in purity) for RNA-seq shown in the revised Figure EV2F.

Although the male gametocytes appear to be normal beyond their gross functional characterization at the level of their ability to generate ookinetes with wt female gametes. Transcriptome would be interesting.

Response:

In this study, we found that AP2-O3 is restrictedly expressed in female gametocytes but not male gametocytes, and *ap2-o3* disrupted parasites developed normal male gametocytes and male gametes but failed to develop mature fertile female gametes. Furthermore, parasite genetic cross experiments also confirmed that the fertile male gametes could be produced in the *ap2-o3* disrupted parasites. Therefore, we focus on the effect of *ap2-o3* disruption on the change of female gametocyte transcriptome.

Whether similar kind of transcription repressor function in male gametocytes waits for further investigation.

Figure 5 - all the male-specific genes chosen for demonstration of expression in "female" gametocytes of the AP2-O3ko background are nuclear - is this exclusively the case? It would appear not to be so are the cytoplasmic protein encoding transcripts not translated? It is reported that male genes are not as abundant as transcripts in O3ko females but have the most abundant male transcripts from these females been tested for expression? It would be useful to have proceeded down the list of abundance rather than just picking some genes which in the absence of a rationale seems to have been what happened.

Response:

We didn't choose the genes according to nuclear or cytosol localization of proteins. After confirming mRNA upregulation of 26 male genes implicated in DNA replication and DNA repair pathways using both RNA-seq and qRT-PCR, four (*dpod2*, *dpod1*, *rpa1*, and *mcm7*) out of these genes were chosen for analyzing their expression change in protein level. We chose these 4 genes (*dpod2*, *dpod1*, *rpa1*, and *mcm7*) because they displayed variable increase in the mRNA level due to AP2-O3 disruption shown in the revised Figure 3E and 3F (red arrow in the images below). As expected, these four proteins are localized in the nucleus. Importantly, these 4 genes displayed similar extent of increase in both mRNA and protein levels in female gametocytes of the AP2-O3 deficient parasites compared to WT parasites (revised

Figure 3).

The model presented in Figure S9 doesn't really summarize the data. In the absence of O3 then there is a superabundance of male transcripts as well as a change in the stoichiometry of the constituency of the female transcriptome. Failure to generate a functional infective ookinete could well be due to male protein subversion of female protein function as the changes to the female transcriptome per se. The model and surrounding discussion should be reworked to more accurately reflect all the different possible speculations.

Response:

Thanks for reviewer's suggestion. I have added the possibility (male protein subversion of female protein function) in the revised discussion and in the proposed model of Figure EV5.

Minor comments

General issues with the use of the definite and indefinite pronouns and syntax which can be sorted with editing.

Response:

We had the manuscript edited by a professional person. The changes in the text are highlighted with red letters.

ALL the references should be checked as many are not listed correctly: volume and page numbers often seem to be missing

Response:

We are very sorry for our incorrect citation. We have checked all the citation of the references and made correction. The changes in the citation are highlighted with red letters.

L 13: "distinct repertoires"

Response: Corrected.

L72: "dogma" is not the correct term

Response:

Changed "but failed to uncover the dogma how the sex-specific gene expression is controlled" to "but failed to uncover how the sex-specific gene expression is controlled".

L134. The sentence beginning on that line does not make sense - pop a verb in there.

Response: Corrected.

L153 Ref 24 does not speak to translational repression of P28 - please find an

appropriate reference

Response:

We apologize for this mistake. We cited the right reference (Mair *et al.* 2006, PMID: 16888139, and Mair *et al.* 2010, PMID: 20169188) for translational repression of P28.

L155: please quantitate the reduced abundance of P28 as it does not appear to be markedly once control loading is taken into account.

Response:

We added quantitative information for blots in the legend and revised Fig 2B.

L158: There is no mention of the word zygote in the cited reference!! 25 Frenal et al. Use the literature accurately!!! What is the correct reference!

Response:

We apologize for this mistake. We cited the right reference (Wang *et al.* 2020, PMID: 32395856) for GAP45 apical localization in zygote after fertilization.

L206 - sentence beginning here needs attention.

Response: Corrected.

Fig 1A Can the authors comment on the apparent expression of AP2-O3 in oocyst sporozoites? This seems to have been ignored in the description of the results.

Response:

Besides at female gametocytes, it is interesting that AP2-O3 is also expressed at mature oocysts but not at developing oocysts in our observations. It is known that large amount of DNA replication occurs during oocyst developing. We speculate that AP2-O3 may play a role like a “brake” for inhibiting genome replication within the

mature oocyst, but not in the developing oocysts, which is consistent with its transcription repressor of genes implicated in DNA replication and DNA repair.

Fig 1B, the scale bars on this CANNOT be uniformly 5µm.

Response:

Please notice that the length of scale bar line (5 µm) in ookinete, oocyst, and sporozoite are different in the revised Fig EV1A and 1B (image below).

1C. spelling of ookinete

Response: Corrected.

Fig 2E-H labels on the x axis?

Response:

The names of these 4 parasite strains (17XNL, $\Delta ap2-o3$, $ap2-o3::6HA$, and $ap2-o3::6HA;\Delta ap2-o3$) are too long to be presented in the X-axis of the revised Figure 1E-H. Instead, we used different color to indicate the strains. To make a clearer

presentation, these 4 different strains were numbered (1, 2, 3, and 4) in the revised Figure 1A. Accordingly, we added the number on the x axis in the revised Figures 1C, E-H. These information was added in the legend of revised Figures 1C, E-H.

FicS4G - Identify the samples clearly - this is just an analysis of purified female gametocytes, correct?

Response:

Yes, we only purified the female gametocytes for transcriptome analysis. Sorry for the unclear interpretation in the manuscript which confused the reviewers. We added the sample information (only female gametocytes) in the revised Figure EV2E.

Referee #2:

The manuscript by Li et al entitled 'Plasmodium transcription repressor AP2-O3 regulates sex-specific identity of gene expression in female gametocytes' reports a functional study of transcription factor AP2-O3, which belongs to the repressor gene ApiAP2 Transcription Factors family in Plasmodium. The study convincingly shows that AP2-O3 is specifically localized in female gametocytes by IFA studies using HA epitope tagged AP2-O3 in *P. yoelli* parasites, and the transcriptional role for AP2-O3 is revealed by immunoblotting and in vitro exflagellation assays using AP2-O3-deficient *P. yoelli* parasites. This revealed that AP2-O3 is essential for ookinete formation and that male gametocytes deficient of AP2-O3 fail to develop into fertile male gametocytes.

In the second part of the study large scale expression studies are reported. RNA-Seq analysis is performed for gaining insights into changes in transcript levels in female gametocytes in AP2-O3 KO parasites compared to wild type female gametocytes. ChIP-seq analysis identified an eight-base DNA binding sequence as promoter sequence for specific male genes.

I do have a few points of concern for the second part of the study for the authors to be addressed:

1) Paragraph 'Transcriptome analysis of female gametocytes' (lines 175-200).

I find this paragraph slightly confusing in the way transcriptome analysis results are presented. It is rather difficult to understand which comparative data analysis is presented. The study has acquired transcriptome data for 'wild type' DFsc7 male and female P.yoelli parasites, and mutant DFsc7; Δ ap2-o3 male and female P.yoelli parasites, but does not present expression results for all the acquired data. I think it would be easier to understand when all data is presented first, and how many statistically relevant upregulated transcripts are determined in female WT DFsc7 parasites compared to male WT DFsc7 parasites and vice versa? The same for male and female DFsc7; Δ ap2-o3 parasites.

Response:

We are sorry for the unclear interpretation in the manuscript. In this study, only female gametocytes (but not male gametocytes) were sorted out by FACS based on a double fluorescent reporter line *DFsc7* we developed (Liu *et al.* 2018, PMID: 30040976). We obtained the transcriptome data for only female gametocytes of *DFsc7* (as wildtype) and *DFsc7; Δ ap2-o3* (as mutant) parasites. We revised the Fig EV2 and added more information for clear interpretation.

The study only reports the identification of 1141 up-regulated genes and 136

down-regulated genes (> 2-fold) in the female gametocytes of *DFsc7;Δap2-o3* compared to WT *DFsc7* parasites. How do these differentially expressed transcripts in *DFsc7;Δap2-o3* parasites compare to expression in WT male parasites. Are the 1141 up-regulated genes in *DFsc7;Δap2-o3* female gametocytes expressed in higher levels in WT male gametocytes compared to WT female gametocytes? This would demonstrate a male specific function as reported in paragraph 'AP2-O3 represses male gene transcription in female gametocytes' (line 203-217) that these up-regulated genes are involved in male specific processes such as DNA replication, so I would expect these genes to be upregulated in WT male gametocytes relative to WT female gametocytes. This can be verified from the acquired transcriptome data.

Response:

Sorry again for the unclear interpretation which confused the reviewers. In this study, only female gametocytes (but not male gametocytes) were sorted out by FACS based on a double fluorescent reporter line *DFsc7*. We obtained the transcriptome for only female gametocytes of *DFsc7* (as wildtype) and *DFsc7;Δap2-o3* (as mutant) parasites.

In the revised Figure 3B, the *P. yoelii* male genes (male-specific or male-preferential) collection used for sex-related comparison of DEGs in this study was from two transcriptome dataset of male gametocytes of rodent malaria parasites, one is a recent sex-gametocyte transcriptome of *P. berghei* (Yeoh *et al.* 2017, PMID:28923023), the other is a male gametocytes transcriptome collected in another project in my lab. In brief, 1584 genes with transcript abundance (FPKM) 3-fold or more in male gametocytes than that in female gametocytes were considered as male genes.

I think it would help when expression data is provided as supplemental tables (excel or txt) with DEGs indicated.

Response:

Thanks for reviewer's suggestion, we added a Dataset EV1 showing the DEGs in this study.

Regarding the RNA Seq data processing procedure, I couldn't find much detail in the

RNA seq Material and Methods section (652-661) about data processing steps, noise level threshold, and data normalisation procedure for female and male gametocyte samples. How many parasites were sorted and how much RNA was extracted from purified male and female parasites? Was a data normalisation procedure applied to RNA seq data from purified male and female parasites? How many transcripts were identified in male and female gametocytes before and after applying a noise level threshold for the RNA seq data?

Response:

We added the detailed information for the RNA-seq and Data analysis in the revised Materials and methods.

Fig S4 shows a cross correlation heatmap of global gene expression between samples of two strains DFsc7 WT and DFsc7; Δ ap2-o3 gametocytes. I understand that a perfect correlation of 1.00 is determined for samples with itself (eg DFsc7 replicate 1 with DFsc7 replicate 1) but it is very unlikely that perfect correlations are observed between replicate samples (eg DFsc7 replicate 1 with DFsc7 replicate 2), and I am wondering if reported correlation coefficients (1.00) for DFsc7 are correct.

Response:

Thank reviewer for pointing this out. We rechecked raw data for the correlation heatmap in revised Fig EV2G. We had the correlation efficiency “1.00” presented in the original data because the format of the decimal position. We presented the updated data in the revised Fig EV2G.

2. Paragraph 'Decreased expression of highly expressed female-specific genes in the AP2-O3 null female gametocytes' (lines 297 - 341).

This paragraph reports differences in gene expression and gene expression profiles supported by figure 7. Gene expression levels are represented as a heatmap with genes sorted by abundance levels in FPKM.

Small point: Which value represent the FPKM signal at left columns from heatmap figures, sum of all samples, or average of all samples, any normalisation applied between samples?

Response:

The value at left columns in heatmap (in revised Fig 6A) is ranked by FPKM average of each gene from three samples replicates within WT. The FPKM values in each row (each gene) were normalized by Z-score normalization. We added the information in the legend of revised Fig 6A.

Expression patterns between samples are difficult to observe in presented heatmaps, and I think clustered heatmaps would provide a much better visual representation of expression profiles between the two parasite strains than the heatmaps in figure 7.

Response:

According to reviewer's suggestion, we generated a clustered heatmap (right image below) showing the expression profiles between WT and KO parasites. In both original heatmap (left image below) and the clustered heatmap (right image below), we can see clearly the up-regulation and down-regulation of genes in the KO over WT, and the number of the up-regulated genes is far more than that of the down-regulated genes. However, only in the left heatmap presenting the genes ranked by expression level (FPKM), it is clear to see that the genes with abundant transcript were down-regulated, while the genes with lower abundance transcript were up-regulated.

Observations are made for differences in expression patterns between abundant transcripts (FPKM>1500) and low abundant transcripts (FPKM<1500). How much of these differences is due to expression level variation in the low abundant transcripts? Are there any DEGs in the low abundant transcript group?

Response:

To address reviewer's concern, we picked up 4589 genes with low abundant transcripts (FPKM<1500) and performed correlation analysis of these genes between WT and KO groups. Importantly, expression correlation analysis (image below) of these genes demonstrated good reproducibility (small difference) among biological replicates within group, but marked difference between WT (*DFsc7*) and KO (*DFsc7;Δap2-o3*) group. Consistent with this result, out of the 1141 upregulated DEGs (in revised Fig 3A), 1140 belong to the low abundant transcripts (FPKM<1500). Together, we believed that the difference (upregulation of the low abundant transcripts) is real phenotypes, but not due to expression level variation in the low abundant transcripts.

The authors observe small RNAseq expression differences for DOZI and CITH genes between wild type and ap2-o3 deficient parasites, while qRT-PCR, IFA and western blotting experiments reveal no significant expression differences at gene and protein level. This indicates to me that the small RNAseq expression differences for DOZI and CITH are not statistically significant and that a statistical analysis of all RNA seq data should be included.

Response:

According to reviewer's suggestion, we added the statistical information for the DOZI/CITH complex-related genes in the revised Appendix Fig S4A (below table). Although we can see statistically significant ($p < 0.05$) increase in these genes mRNA level in KO compared to WT, but the increase extent is quite small with fold change < 2 in all 9 genes, which is consistent with no marked expression difference at protein level.

These observations are also similar as what we observed in the revised Figure 4: only the genes (*dpod2* and *dpod1*) with marked increased transcript level displayed strong upregulated protein expression, but not the genes (*rpa1*, and *mcm7*) with minor increased transcript level.

A

Gene ID	Gene name	Mean±SD FPKM in DFsc7	Mean±SD FPKM in DFsc7;Δap2-o3	p-value	Log2(FC)
PY17X_1220900	dozi	289.9±45.1	401.8±20.9	0.0175	0.47
PY17X_1304900	cith	188.1±19.2	268±32.5	0.0215	0.51
PY17X_0415700	eIF4e	46±4.9	75.4±3.6	0.0011	0.71
PY17X_1441700	pabp1	429.8±62.6	698.3±22.1	0.0022	0.70
PY17X_1035100	celf2	61±6.9	88.3±3.9	0.0039	0.53
PY17X_1425300	alba1	1478.8±190.3	1970.1±95.2	0.0161	0.41
PY17X_1364900	alba2	168.9±7.7	254.8±17.3	0.0014	0.59
PY17X_1207600	alba3	134.2±28.3	215.5±16.6	0.0127	0.68
PY17X_0410900	phosphoglycerate mutase	44.9±4.4	55.7±6.6	0.0791	0.30

3. Paragraph 'Sequential expression of AP2-O3 and AP2-O before and after female gametogenesis' (lines 376-400). This paragraph presents some functionals result for AP2-O, which is another member of the ApiAP2 Transcription Factors. However, I don't see the relevance for these experiments in this study, and reported results may not be of great importance as these are not part of the Discussion.

Response:

We agreed with reviewer's comments. It was why we presented the results at the supplementary data (revised Appendix Fig S5). In this part, we present evidences indicating sequential expression of AP2-O3 and AP2-O before and after female gametogenesis using a doubly tagged strain *ap2-o3::6HA;ap2-o::4Myc*, which may provide better understanding of these two TFs critical in parasite development (female gametocyte-female gamete-zygote-ookinete).

Dear Prof. Yuan,

Thank you for submitting your revised manuscript. It has now been seen by one of the original referees.

I apologize for this unusual delay in getting back to you. It took longer than anticipated to receive the referee report due to the recent holiday season.

As you can see, the referee finds that the study is significantly improved during revision and recommends publication. Before I can accept the manuscript, I need you to address some minor points below:

- The initials in the 'Author Contributions' section should be firstname/surname.
- We note that Zengang Yang is missing from the 'Author Contributions'.
- We notice the phrase 'data not shown' on page 5, which is not allowed as per journal policy. Please either remove the statement or show the data.
- We note that Figure 4I is currently not called out in the manuscript text.
- We note that Appendix Fig S1 panels are currently not called out in the manuscript text.
- Some of the HA staining panels of Figure EV1 A and B appear empty. Please clarify.
- Please make the GSE157456 RNA-seq and GSE157454 ChIP-Seq data publicly available.
- Papers published in EMBO Reports include a 'synopsis' and 'bullet points' to further enhance discoverability. Both are displayed on the html version of the paper and are freely accessible to all readers. The synopsis includes a short standfirst summarizing the study in 1 or 2 sentences that summarize the paper and are provided by the authors and streamlined by the handling editor. I would therefore ask you to include your synopsis blurb and 3-5 bullet points listing the key experimental findings.
- In addition, please provide an image for the synopsis. This image should provide a rapid overview of the question addressed in the study but still needs to be kept fairly modest since the image size cannot exceed 550x400 pixels.
- Our production/data editors have asked you to clarify several points in the figure legends (see attached document). Please incorporate these changes in the attached word document and return it with track changes activated. I am aware that the comments were made on an earlier version of the manuscript, please use the attached document as a reference and perform the changes on the latest version of the text.

Thank you again for giving us to consider your manuscript for EMBO Reports, I look forward to your minor revision.

Kind regards,

Deniz Senyilmaz Tiebe

--

Deniz Senyilmaz Tiebe, PhD
Editor
EMBO Reports

Referee #2:

I would like to thank the authors for addressing adequately my comments.

The authors have addressed all minor editorial requests.

Dear Prof. Yuan,

Thank you for submitting your revised manuscript. I have now looked at everything and all is fine. Therefore, I am very pleased to accept your manuscript for publication in EMBO Reports.

Congratulations on a nice study!

Kind regards,

Deniz Senyilmaz Tiebe

--

Deniz Senyilmaz Tiebe, PhD

Editor

EMBO Reports

--

At the end of this email I include important information about how to proceed. Please ensure that you take the time to read the information and complete and return the necessary forms to allow us to publish your manuscript as quickly as possible.

As part of the EMBO publication's Transparent Editorial Process, EMBO reports publishes online a Review Process File to accompany accepted manuscripts. As you are aware, this File will be published in conjunction with your paper and will include the referee reports, your point-by-point response and all pertinent correspondence relating to the manuscript.

If you do NOT want this File to be published, please inform the editorial office within 2 days, if you have not done so already, otherwise the File will be published by default [contact: emboreports@embo.org]. If you do opt out, the Review Process File link will point to the following statement: "No Review Process File is available with this article, as the authors have chosen not to make the review process public in this case."

Should you be planning a Press Release on your article, please get in contact with emboreports@wiley.com as early as possible, in order to coordinate publication and release dates.

Thank you again for your contribution to EMBO reports and congratulations on a successful publication. Please consider us again in the future for your most exciting work.

THINGS TO DO NOW:

You will receive proofs by e-mail approximately 2-3 weeks after all relevant files have been sent to our Production Office; you should return your corrections within 2 days of receiving the proofs.

Please inform us if there is likely to be any difficulty in reaching you at the above address at that

time. Failure to meet our deadlines may result in a delay of publication, or publication without your corrections.

All further communications concerning your paper should quote reference number EMBOR-2020-51660V3 and be addressed to emboreports@wiley.com.

Should you be planning a Press Release on your article, please get in contact with emboreports@wiley.com as early as possible, in order to coordinate publication and release dates.

Corresponding Author Name: Jing Yuan

Manuscript Number: EMBOR-2020-51660V3